# On Logic-based Self-Explainable Graph Neural Networks

**Alessio Ragno**
INSA Lyon, CNRS, LIRIS UMR 5205,
F-69621 Villeurbanne, France
`alessio.ragno@insa-lyon.fr`

**Marc Plantevit**
EPITA Research Laboratory (LRE),
F-94276, Le Kremlin-Bicêtre, France
`marc.plantevit@epita.fr`

**Céline Robardet**
INSA Lyon, CNRS, LIRIS UMR 5205,
F-69621 Villeurbanne, France
`celine.robardet@insa-lyon.fr`

## Abstract

Graphs are complex, non-Euclidean structures that require specialized models, such as Graph Neural Networks (GNNs), Graph Transformers, or kernel-based approaches, to effectively capture their relational patterns. This inherent complexity makes explaining GNNs decisions particularly challenging. Most existing explainable AI (XAI) methods for GNNs focus on identifying influential nodes or extracting subgraphs that highlight relevant motifs. However, these approaches often fall short of clarifying how such elements contribute to the final prediction. To overcome this limitation, logic-based explanations aim to derive explicit logical rules that reflect the model's decision-making process. Current logic-based methods are limited to post-hoc analyzes and are predominantly applied to graph classification, leaving a significant gap in intrinsically explainable GNN architectures. In this paper, we explore the potential of integrating logic reasoning directly into graph learning. We introduce LogiX-GIN, a novel, self-explainable GNN architecture that incorporates logic layers to produce interpretable logical rules as part of the learning process. Unlike post-hoc methods, LogiX-GIN provides faithful, transparent, and inherently interpretable explanations aligned with the model's internal computations. We evaluate LogiX-GIN across several graph-based tasks and show that it achieves competitive predictive performance while delivering clear, logic-based insights into its decision-making process.

## 1   Introduction

Graph Neural Networks (GNNs) are powerful models for learning from graph-structured data, excelling in tasks like node classification and graph classification [25, 29]. However, their complex message-passing mechanisms make it difficult to interpret how individual graph components influence predictions, limiting their use in critical domains that require transparency.

Several Explainable AI (XAI) techniques have been proposed to address the opacity of GNNs, typically falling into post-hoc and ante-hoc approaches. Post-hoc methods aim to explain an already trained black-box GNN by identifying influential graph components (e.g., nodes, edges, or subgraphs) that contribute to a specific decision [31, 20, 15]. These methods include perturbation-based approaches, gradient-based attribution, and concept-based methods that aim to extract meaningful graph patterns. Although effective, they often lack fidelity to the internal reasoning of the model, as they hardly capture the internal process of the model and can generate inconsistent or misleading

39th Conference on Neural Information Processing Systems (NeurIPS 2025).

explanations [24]. Ante-hoc, or self-explainable, models, on the other hand, integrate interpretability into the learning process. Self-explainable GNNs (SE-GNNs) aim to ensure that the learned representations are inherently interpretable, often by enforcing constraints on the structure of explanations or by explicitly modeling feature importance. However, existing SE-GNNs present some limitations: topology-based approaches [17, 8, 22, 36, 21] identify the most relevant subgraphs while providing little insight into the logical rules underlying how the model represents graphs and makes its decisions [18, 19]; symbolic approaches propose distillation of the models into simpler or more interpretable architectures such as decision trees; and other approaches restrict the focus on node features, without using information about substructures [5, 3].

To address these limitations, logic-based XAI methods provide an alternative approach by extracting human-interpretable logical rules that govern the model's decision-making process [26]. Unlike gradient-based or perturbation-based explanations, logic rules explicitly define how features interact to produce a given outcome. When applied to GNNs, these approaches aim to map high-dimensional feature spaces to Boolean expressions that describe the presence of specific subgraph motifs or node interactions [2, 1]. However, currently existing logic-based GNN explainers are exclusively post-hoc or based on distillation of the GNNs using decision trees [19].

Recent advances in logic-explained networks for tabular data provide potential directions. Logic Explained Networks (LENs) [6] and Transparent Explainable Logic Layers (TELL) [23] propose architectures convertible into logical rules. Particularly relevant is TELL's approach, where non-negative weight constraints allow direct extraction of first-order logic rules from the model parameters. However, these architectures were designed specifically for tabular data and cannot handle the complex relational structure of graphs. To address the lack of self-explainable logic GNN, in this work, we design a novel architecture that is intrinsically convertible into logic rules.

The theoretical foundation for connecting GNNs with formal logic is established by Barceló et al. [4], who demonstrate that some GNN architectures can represent statements in graded modal logic (GML) [10]. GML extends standard modal logic with numerical constraints, allowing expressions like "at least 3 neighboring nodes satisfy property $P$", a natural fit for graph-based reasoning. While Barceló et al. [4] prove that GNNs can express GML propositions, it does not address the reverse problem: architecting GNNs to explicitly yield interpretable GML rules.

Constraining GNNs to adhere to logic comes with different challenges. To implement our model, LogiX-GIN, we modify the GIN architecture [30], such that it is possible to extract explanations in the form of GML. The choice of the GIN layer is dictated by its aggregation scheme, which uses the sum over neighboring nodes. This aggregation allows us to respect the counting functionality of GML. However, in order to count over input nodes, we cannot use a simple threshold as is done in TELL. For this reason, we introduce a preprocessing function that binarizes the input features depending on learnable intervals. An additional challenge comes from the use of sigmoid activations directly on the weights, as proposed in TELL. These tend to hinder gradient flow in multi-layer settings. This limits the effective training of models with many layers, as early layers receive little to no gradient updates. For this reason, we propose to first train a black-box model, GIN, and use it to pretrain our LogiX-GIN model. This allows for pretraining each layer separately and therefore avoids problems such as vanishing gradients.

Overall, the main contributions of our work are as follows: we propose LogiX-GIN, the first graph neural network that integrates logical reasoning directly into its architecture; we introduce a novel preprocessing function for automatic binarization of input features using intervals; we propose the first implementation of multi layer logic-based neural architecture by applying a distillation-based pretraining strategy to address vanishing gradient issues; we conduct extensive experiments to assess the interpretability, predictive performance, and limitations of LogiX-GIN; we provide an open-source implementation of our proposed approach and the conducted experiments[1].

The remainder of this work is structured as follows: we begin by surveying the current literature on XAI for GNNs, self-explainable models and logic-based XAI; subsequently, we detail the mathematical foundations needed to comprehend our approach; we then present LogiX-GIN; we follow with an experimental evaluation on several datasets; finally, we summarize our findings and suggest potential directions for future research.

---

[1]Public GitHub repository: `https://github.com/spideralessio/LogiX-GIN`

## 2  Related Work

In this section, we present the related literature to our work. We start by introducing self-explainable models for GNNs, and we then continue focusing on approaches that deliver explanations in the form of logic. Finally, we focus on approaches aimed at delivering logic explanations for GNNs.

**Self-Explainable GNNs.**   SE-GNNs are designed to generate intrinsic explanations during inference, removing the need for post-hoc interpretation methods. A central approach in this paradigm involves the use of prototypes, intended as representative training instances, to construct inherently interpretable models. Prototype-based SE-GNNs adapt prototypical mechanisms to the graph domain, where the output class is determined using the similarity between the input and prototypes [36, 22, 21]. Despite the interpretability of the case-based approach, these models rely on embedding similarity, which isn't guaranteed to actually reflect structural similarity. Other approaches leverage information bottleneck methods to identify maximally informative subgraphs, ensuring that the extracted explanations retain only the most relevant structural components. For example, GIB [32] and VGIB [33] utilize information-theoretic principles to compress node and edge representations while preserving predictive performance. Kernel-based approaches such as KerGNNs [13] take a different route by employing graph kernels to provide explanations through kernel activations, allowing for a structured decomposition of the learned representations.

**Logic-based XAI.**   Logic-based explainability methods [9] aim to improve model interpretability by incorporating logical constraints or symbolic reasoning directly into the learning process. Traditional machine learning models, like decision trees and rule-based classifiers, inherently provide logical explanations by explicitly representing decision boundaries. Although these models are transparent, they typically struggle with complex, high-dimensional data and cannot be easily integrated into deep learning frameworks due to their non-differentiability, which prevents the use of gradient-based optimization. This limitation has led to the development of neural architectures that incorporate logical reasoning while remaining compatible with backpropagation. A prominent example is represented by LENs [6], a class of neural networks that can be interpreted through logical rules. LENs are built on binary inputs and employ strong regularization, allowing the construction of truth tables from which logical rules can be derived. However, due to the post-hoc truth table procedure, these rules may not always faithfully represent the model's true behavior [23]. To overcome this issue, Ragno et al. [23] introduce TELL. TELL is an architecture with positive-weight constraints that ensure direct and faithful translation into logical rules. Furthermore, TELL is capable of handling continuous input data through a preprocessing function that automatically learns thresholds over features.

**Logic-based Explanations for GNNs.**   Among the first logic-based approaches for GNNs, GLGExplainer [2] proposes to extract logical rules over explanations generated by another post-hoc method, such as PGExplainer [15]. This is achieved through the application of LENs over concept activation vector that encode the presence of specific subgraphs. While innovative, this approach relies on the application of an instance-level post-hoc procedure and on LENs, that were demonstrated to not be properly faithful [23, 1]. With GraphTrail, Armgaan et al. [1] propose to overcome these issues by using computation trees to identify relevant subgraphs and avoiding the need of post-hoc methods. These approaches primarily aim to identify rules over subgraphs in order to gain insight into the model's behavior. Taking a different direction, Pluska et al. [19] propose distilling a symbolic model from a GNN using iterative decision trees. Despite this difference, their method is also post-hoc, as the rules are extracted from graph embeddings. Additionally, the final model is no longer a GNN and cannot be reused in other neural network settings, thus losing the advantages of neural representations.

Although we share with the aforementioned works the goal of incorporating logic-based reasoning, our approach differs from current literature in a key aspect: rather than extracting rules in a post-hoc fashion, we aim to directly design a GNN model that is self-interpretable by construction. For this reason, in our case, distillation is used solely as a means to pre-initialize the self-interpretable model, not as an end in itself.

# 3 Background

We now introduce the key concepts and notations underlying our approach, focusing on how each element supports our logic-based self-explainable GNN. We start by formalizing GNNs, the base model we build upon. We then show how architectural constraints can enable direct extraction of logic rules from neural layers, introducing the core ideas of TELL. Finally, we present the mathematical foundations of GML, the formal system we use to express the logic rules produced by our model.

**Graph Neural Networks.**  GNNs are neural networks designed for graph-structured data. They generally operate via message passing, where at each layer $k$ a node $v$ aggregates messages from its neighbors and updates its representation. Formally, this process can be written as:

$$h_v^{(k)} = \phi^{(k)} \left( h_v^{(k-1)}, \mu^{(k)} \left( \{ h_u^{(k-1)} : u \in \mathcal{N}(v) \} \right) \right), \tag{1}$$

Here, $h_v^{(k)}$ denotes the updated representation of node $v$ at layer $k$, using the ones at layer $k-1$, $h_v^{(k-1)}$. In the first layer, $h_v^{(k-1)}$ directly correspond to node features. The message function $\mu^{(k)}$ aggregates information from the neighborhood, and the update function $\phi^{(k)}$ combines it with the node's current state. Since nodes in a graph have no inherent ordering, the aggregation function must be permutation-invariant.

We use GIN [30] model, where the aggregation function is the sum, and the update function is a multi-layer perceptron (MLP) as this combination has been theoretically shown to achieve maximum discriminative power among message-passing GNNs. The sum aggregation can distinguish between different neighborhood structures and the MLP update function provides the necessary expressive capacity to obtain meaningful representations, enabling GIN to approximate the 1-WL (Weisfeiler-Lehman) graph isomorphism test, which is a strong baseline for distinguishing non-isomorphic graphs. In this architecture, each layer computes:

$$h_v^{(k)} = \text{MLP}^{(k)} \left( (1 + \epsilon^{(k)}) h_v^{(k-1)} + \sum_{u \in \mathcal{N}(v)} h_u^{(k-1)} \right). \tag{2}$$

with $\epsilon^{(k)}$ a learnable parameter. Multiple layers are stacked to capture increasingly abstract features. For node-level tasks, the resulting node embeddings are directly used. For graph-level tasks, a readout function (e.g., sum, mean, or max) aggregates node embeddings into a global representation for classification.

**Transparent Explainable Logic Layer.**  TELL is a neural component designed to be interpretable through a direct conversion into first-order logic formulas. It constrains a standard linear layer, with $I$ input neurons and $O$ output ones, by enforcing non-negative weights and applying a threshold-based activation. Given an input $X \in \mathbb{R}^I$, it computes the output $y = \sigma(XW^+ + b) \in \mathbb{R}^O$. $W^+ \in \mathbb{R}_{\geq 0}^{I \times O}$ is a weight matrix constrained to have only non-negative entries, $b \in \mathbb{R}^O$ is a bias term, and $\sigma$ is the sigmoid activation function.

A key feature of TELL is that it allows to be interpreted through logic rules. Considering the j-th binarized output $y^{\text{bin}}[j] = \mathbb{1}_{y[j]>0.5}$, it can then be associated with a rule in disjunctive normal form (DNF): $y^{\text{bin}}[j] = 1 \iff \sum_{i \in S} W^+[i,j] > -b[j]$. $S$ denotes the minimal subsets w.r.t set inclusion of the input sufficient to activate the neuron $y[j]$ [23]. Following this, $E_j = \bigvee_{S \in \mathcal{S}_j} \bigwedge_{i \in S} x_i$ can be defined as the logic rule for $y[j]$, where $\mathcal{S}_j$ contains all the minimal subsets of inputs that satisfy the threshold $-b[j]$. This ensures a direct and interpretable mapping from network parameters to logical expressions. To extend TELL to handle continuous inputs, a preprocessing binarization step introduces an adaptive thresholding mechanism. This is done using $\tilde{X} = \sigma(X \odot \exp(\tilde{W}) + \tilde{b})$, with $\odot$ representing element-wise multiplication, and $\tilde{W} \in \mathbb{R}^I$ and $\tilde{b} \in \mathbb{R}^I$ being learnable parameters. Each transformed feature can then be interpreted as a binary predicate where $\tilde{x}[i] = \mathbb{1}_{x[i] > -\frac{\tilde{b}[i]}{\exp(\tilde{W}[i])}}$.

These binary features can be incorporated into logical expressions, preserving TELL's explainability while enabling it to process real-valued data effectively.

**Graded Modal Logic.** GML is a fragment of first-order logic (FO) designed to express local properties of nodes in relational structures such as graphs, with support for counting. It restricts quantification to the immediate neighborhood of a node and ensures that formulas are evaluated locally, which makes it well-suited for graph-structured data.

Let us consider a first-order language that includes a symbol $E(x, y)$, for a binary relation between representing adjacency between nodes $x$ and $y$ of an undirected graph, and a finite collection of unary predicate symbols $\{P_1(x), \ldots, P_I(x)\}$ of size $I$, which represent properties, features or types associated with individual nodes.

A graph structure $G = (V, E, \{P_i\}_{i \in I})$ consists of: a finite set of nodes $V$; a symmetric edge relation $E \subseteq V \times V$; and, a collection of unary relations $P_i \subseteq V$ interpreting each predicate symbol.

Formulas of GML are a syntactic restriction of FO with exactly one free variable $x$. The set of formulas $\varphi(x)$ is defined inductively by:

$$\varphi(x) ::= P(x) \mid \neg\psi(x) \mid \psi_1(x) \wedge \psi_2(x) \mid \psi_1(x) \vee \psi_2(x) \mid \exists^{\geq N} y \, (E(x, y) \wedge \psi_3(y)), \quad (3)$$

where $N \in \mathbb{N}_{>0}$, and $\psi(x), \psi_1(x), \psi_2(x), \psi_3(y)$ are GML formulas.

Given a graph $G$ and a node $v$, the relation $G, v \models \varphi(x)$ that indicates that $v \in G$ satisfies a generic GML formula $\varphi(x)$ is defined as follows:

- $G, v \models P(x)$: node $v$ satisfies the atomic predicate $P$ (i.e., $v$ belongs to the unary predicate $P$),
- $G, v \models \neg\psi(x)$: node $v$ does not satisfy the formula $\psi(x)$ (negation),
- $G, v \models \psi_1(x) \wedge \psi_2(x)$: node $v$ satisfies both subformulas $\psi_1(x)$ and $\psi_2(x)$ (logical conjunction),
- $G, v \models \psi_1(x) \vee \psi_2(x)$: node $v$ satisfies at least one of the subformulas $\psi_1(x)$ or $\psi_2(x)$ (logical disjunction),
- $G, v \models \exists^{\geq N} y \, (E(x, y) \wedge \psi_3(y))$: node $v$ has at least $N$ neighbors $u$ such that $(v, u) \in E$ and each $u$ satisfies $\psi_3(y)$ (quantified neighborhood condition).

A distinctive feature of GML is that all quantification is conditioned by the adjacency predicate $E(x, y)$. This ensures that the evaluation of any formula at a node $v$ depends only on $v$ and its neighborhood, not on the entire graph. Consequently, the logic is strictly less expressive than full FO, but computationally more tractable and naturally suited for expressing local patterns in graphs. For concrete examples of GML formulas, we refer the reader to Appendix B. The correspondence between GML and local properties of graphs plays a central role in recent work by Barceló et al. [4], where the authors show that it precisely characterizes the class of node properties computable by a certain class of graph neural networks.

## 4 LogiX-GIN

In this section, we formalize LogiX-GIN. We start by defining the different components of the architecture and then we present implementation and training details.

**LogiX-GIN components.** Like the original GIN architecture, LogiX-GIN performs aggregates embedding values through the sum, then a learnable parameterized function $\beta$ binarizes the aggregated values in order to apply a logic transformation $\lambda$ that is responsible to learn logic rules (as the one in TELL). The overall operation of the $k$-th LogiX-GIN layer can be formulated with the notation of Equation 1 as follows:

$$h_v^{(k)} = \lambda^{(k)} \left( \beta^{(k)} \left( \sum_{u \in \mathcal{N}(v) \cup \{v\}} h_u^{(k-1)} \right) \right). \quad (4)$$

The first step involves aggregating the information of node $v \in \mathcal{V}$ and its neighbors by summing their features: $a_v^{(k)} = \sum_{u \in \mathcal{N}(v) \cup \{v\}} h_u^{(k-1)}$. This aggregation is equivalent to the one of GIN when setting $\epsilon = 0$, which is necessary to represent counting operations in GML. After summing binary

embeddings from neighboring nodes, the aggregated features become integer-valued. To apply logic-based rules, we binarize them using a novel thresholding mechanism. Specifically, we need a function capable to activate if its input falls within certain intervals. For this reason, we design a parametric Fourier step function, which flexibly and differentiably activates (outputs one) when counts fall within learnable value ranges:

$$\tilde{a}_v^{(k)} = \beta^{(k)}(a_v^{(k)}) = \text{clamp}\left(\frac{1}{2}\left(\frac{\sum\limits_{i=0}^{\tau} \frac{\sin((2i+1)(a_v^{(k)} \odot \tilde{W}^{(k)} + \tilde{b}^{(k)}))}{2i+1}}{\sum\limits_{i=0}^{\tau} \frac{\sin\left((2i+1)\frac{\pi}{2}\right)}{2i+1}} + 1\right), 0, 1\right), \qquad (5)$$

where $\odot$ denotes element-wise multiplication (Hadamard product), $\tau$ is a temperature hyperparameter (we use $\tau = 10$), $\tilde{W}^{(k)} \in \mathbb{R}_{>0}^d$ and $\tilde{b}^{(k)} \in \mathbb{R}_{>0}^d$ are learnable parameters, and $d$ the hidden dimension. $\beta$ is a periodic step function with respect to $a_v^{(k)}$, where $\tilde{W}^{(k)}$ and $\tilde{b}^{(k)}$ modulate the width and the displacement of the counting intervals corresponding to an activation. Specifically, the intervals can be obtained as follows:

$$\tilde{a}[i]_v^{(k)} = 1 \iff a[i]_v^{(k)} \in \bigcup_{n \in \mathbb{Z}} \left[\frac{2\pi n - \tilde{b}[i]^{(k)}}{\tilde{W}[i]^{(k)}}, \frac{(2\pi n + \pi) - \tilde{b}[i]^{(k)}}{\tilde{W}[i]^{(k)}}\right] = \mathcal{I}_{\beta^{(k)}, i}. \qquad (6)$$

This enables the extraction of literals that are active only under precise counting conditions. To better clarify the functioning of $\beta$, we provide in Appendix A an example of the function varying the parameters $\tilde{W}^{(k)}$ and $\tilde{b}^{(k)}$.

The final transformation $\lambda^{(k)}$ consists of the constrained linear transformation followed by a sigmoid activation function, as proposed in TELL:

$$h_v^{(k)} = \lambda^{(k)}\left(\tilde{a}_v^{(k)}\right) = \sigma\left(\frac{\tilde{a}_v^{(k)} W^{+(k)} + b^{(k)}}{\tau}\right), \quad \text{with } \tau = 10^{-4} \qquad (7)$$

Using the results of Ragno et al. [23], we use non-negative weights $W^{+(k)}$ in order to constrain the model to be monotonic. This allows for retrieving logic rules in FO over the activations $\tilde{a}_v^{(k)}$, which correspond to formulas in GML over the literals $h_u^{(k-1)}$ of each node $u \in \mathcal{V}$. Such logic rules are identified by the sets of weights of $W^{+(k)}$, which sum up to a value greater than $-b^{(k)}$. For a more detailed and formal demonstration, the reader can refer to Appendix D.

**LogiX-GIN overview.** Here, we describe the practical implementation details of the LogiX-GIN architecture used in our experiments. The model is composed of five sequential LogiX-GIN layers, whose outputs are concatenated and passed through three global readout functions: sum, mean, and max. These aggregation functions extract global graph-level features by computing the total, average, and maximum activations of the rules learned by the five graph layers. The resulting features are then processed by a final TELL layer, which learns first-order logic rules over these activations to perform the prediction.

As previously discussed, the presence of sigmoid activation functions within each LogiX-GIN layer introduces significant challenges when training deep architectures. To address this, we adopt a knowledge distillation strategy for pretraining. Specifically, we initialize an auxiliary model with the same architecture, but replace the LogiX-GIN layers with standard GIN layers. To encourage binarized representations, this GIN model uses Gumbel-sigmoid activations. After training the GIN model, we use it as a teacher to guide the LogiX-GIN model for the first half of the training process: we minimize the cross-entropy loss between the hidden states of the GIN and LogiX-GIN models at each layer. This yields a layer-wise pretraining of the LogiX-GIN model. In the second half of training, we finetune the full LogiX-GIN model using standard supervised learning. Throughout the entire training procedure, optimization is carried out using gradient descent to minimize the cross-entropy loss combined with an $L_1$ regularization term on the weights $W^+$.

Since there is no built-in constraint on the size of the learned logic rules, the resulting model can sometimes produce overly complex and less interpretable explanations. To enhance interpretability, we apply a post-training pruning procedure. This step consists of a final optimization stage in which each layer is trained individually in reverse order (from the last to the first) using an additional Hoyer regularization term: $\mathcal{L}_{\text{Hoyer}} = \frac{\|W^+\|_1}{\|W^+\|_2 + \delta}$, where $\delta$ is a small constant to avoid division by zero. This regularization promotes sparsity in a structured way: unlike standard $L_1$ regularization, the Hoyer

loss encourages the use of a few large weights while pushing the remaining ones toward zero. Since the size of a logic rule is determined by how many weights need to be summed to reach the threshold $-b$, this sparsity directly reduces rule complexity. By applying this pruning procedure gradually and layer-wise, we minimize any potential impact on model performance while significantly improving the readability and compactness of the learned rules.

LogiX-GIN can provide explanations in different forms: logic rules can be extracted for each single layer to obtain a fine-grained set of rules that exactly reflect the model behavior; for a simpler overview, global logic rules can be obtained by identifying motifs that when present activate class-specific rules of the last layer; node attributions visualize node level contributions towards a prediction. Further details on the LogiX-GIN pipeline are detailed in Appendix C.

## 5 Experiments

In this section, we perform extensive experiments to evaluate our proposal. We begin by comparing the classification performances of LogiX-GIN with respect to its black-box counterpart. Successively, we perform an analysis of the interpretability of LogiX-GIN also in comparison with state-of-the-art approaches. We mainly focus on a logic-based analysis by identifying the global logic rules and layer-wise rules. For an additional analysis, we also report an experiment on node attributions in the Appendix E, comparing with other self-interpretable models and post-hoc approaches.

As the GIN layer, which inspires this work, is mainly developed for graph classification, we focus on this task for the experimental section. However, to offer a broader overview of the capabilities of LogiX-GIN, we also test the accuracy performances on some node classification datasets. For graph classification, the black-box GNNs are composed of 5 GIN layers, followed by a max, a sum and mean readouts. The readouts are concatenated and finally fed into an 2-layer MLP. We perform our experiments on the following 7 graph and 3 node classification datasets spanning synthetic graphs (BA2Motifs [15], BAMultiShapes [2], BAShapes, BACommunity and TreeGrid [31]), molecular graphs (MUTAG [11], Mutagenicity [14], NCI1 [28], and BBBP [16]), and protein graphs (PROTEINS [12]). We use grid-search to find the optimal hyperparameter combinations and report the test set results of the best performing hyperparameters on the validation set. Specifically, to ensure statistical validity, we report mean and standard deviation values over 10 different seeds.

In Table 1, we report the accuracy of the LogiX-GIN model compared to that of the black-box model across all datasets. In all cases, LogiX-GIN achieves accuracy levels within 3% of the black-box model, demonstrating strong performance despite the addition of logic constraints. These results indicate that incorporating logic constraints has a limited impact on predictive accuracy overall. While some trade-off in performance is expected, the primary goal of this work is to explore the integration of logic for enhanced model transparency. Table 2, instead, reports the accuracy scores of several SE-GNNs on graph classification. We observe that, compared to other self-interpretable approaches, our method reaches state-of-the-art performances on 5 datasets out of 7. In the remainder

Table 1: Accuracy on Graph and Node Classification. Values are reported as $\mu \pm \sigma$ over 10 seeds. We report the accuracy of the black-box model and LogiX-GIN. For an easier comparison, the column "Acc. Decay" indicates the difference between the mean accuracy of the black-box and the LogiX-GIN.

| Dataset | Black-box GIN | LogiX-GIN | Acc. Decay | Class. Obj. |
|---|---|---|---|---|
| BAMultiShapes | $100.00 \pm 0.00$ | $100.00 \pm 0.00$ | 0.00 | Graph |
| BA2Motifs | $100.00 \pm 0.00$ | $100.00 \pm 0.00$ | 0.00 | Graph |
| BBBP | $87.95 \pm 2.07$ | $85.90 \pm 0.99$ | 2.05 | Graph |
| MUTAG | $84.74 \pm 8.02$ | $82.63 \pm 5.58$ | 2.11 | Graph |
| NCI1 | $81.87 \pm 1.39$ | $78.93 \pm 1.51$ | 2.94 | Graph |
| PROTEINS | $72.59 \pm 3.04$ | $72.05 \pm 5.77$ | 0.54 | Graph |
| Mutagenicity | $82.21 \pm 1.87$ | $79.31 \pm 1.25$ | 2.90 | Graph |
| BaShapes | $97.14 \pm 0.95$ | $94.29 \pm 2.52$ | 2.85 | Node |
| BaCommunity | $81.00 \pm 1.59$ | $84.14 \pm 4.55$ | -3.14 | Node |
| TreeGrid | $100.00 \pm 0.00$ | $98.71 \pm 0.87$ | 1.29 | Node |

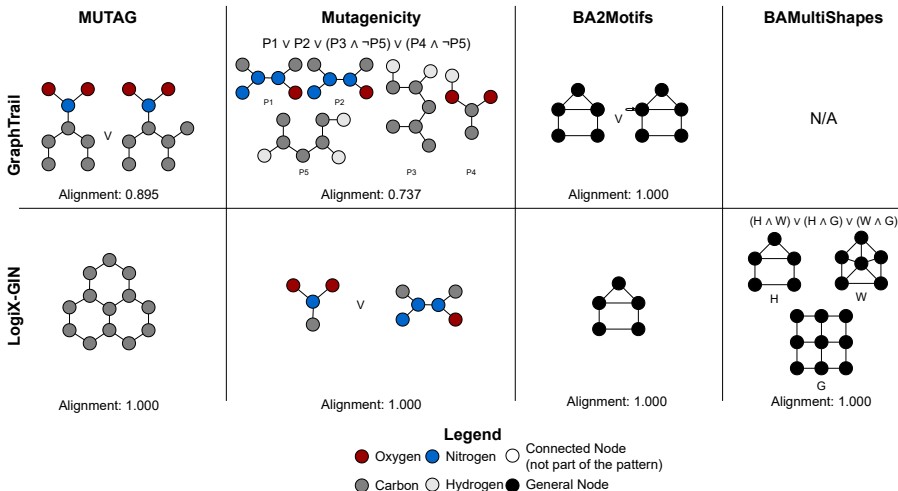

Figure 1: Visual comparison between rules extracted with GraphTrail and LogiX-GIN. Alignment is measured as the accuracy between the predictions of the model and the logic activations of the rules following Armgaan et al. [1]. Additional details about the explanations extraction are provided in Appendix C.

of this section, we focus on the interpretability of LogiX-GIN, while in Appendix I, J and H, we detail hyperparameters settings and perform ablation studies.

**Obtaining faithful global logic rules.** Here, we compare the explanations produced by LogiX-GIN with those of GraphTrail [1]. As discussed in previous sections, despite sharing a logical foundation, the two methods differ substantially in their explanatory procedures: GraphTrail is a post-hoc method that generates rules over computation trees, whereas LogiX-GIN embeds logical reasoning directly into its architecture. LogiX-GIN, in fact, employs GML as its reasoning logic, which goes beyond simply detecting the presence of specific patterns via first-order logic. Despite these significant differences, if we narrow our focus to LogiX-GIN's final layer, the activations of the literals after the readout function can be interpreted as indicative patterns whose presence directly affects the final prediction. This observation forms the basis for our comparison between LogiX-GIN and GraphTrail.

For this experiment, we employ the four datasets used in the study by Armgaan et al. [1]. Due the 5-layer architecture of the GIN blackbox, GraphTrail's SHAP-based computation becomes particularly resource-intensive. For this reason, on the BAMultiShapes dataset, GraphTrail fails to generate explanations within a feasible time frame (2 days). In Figure 1, we illustrate the patterns learned by both models. The LogiX-GIN patterns are extracted by analyzing the activations over the validation set. Full activation data is provided in Appendix F.

Notable differences emerge, especially on MUTAG and Mutagenicity. For MUTAG, GraphTrail identifies the $NO_2$ group, a well-known mutagenic substructure, while LogiX-GIN emphasizes carbon-based structures, especially aromatic atoms, which also play a key role in mutagenicity [1, 11]. However, we also anticipate that a deeper analysis reveals that LogiX-GIN does recognize

Table 2: Comparison of SE-GNNs across graph classification datasets.

| Dataset | LogiX-GIN | PiGNN | GIB | KerGNN | GNAN |
|---|---|---|---|---|---|
| Ba2Motifs | $100.00 \pm 0.00$ | $99.89 \pm 0.31$ | $100.00 \pm 0.00$ | $98.80 \pm 0.75$ | $49.10 \pm 0.66$ |
| BaMultiShapes | $100.00 \pm 0.00$ | $85.40 \pm 5.08$ | $97.60 \pm 2.06$ | $83.20 \pm 1.94$ | $49.10 \pm 0.66$ |
| MUTAG | $82.63 \pm 5.58$ | $82.51 \pm 10.48$ | $90.53 \pm 6.14$ | $82.11 \pm 9.18$ | $55.79 \pm 16.43$ |
| Mutagenicity | $79.31 \pm 1.25$ | $82.39 \pm 1.68$ | $80.14 \pm 0.98$ | $73.32 \pm 2.93$ | $55.36 \pm 0.52$ |
| NCI1 | $78.93 \pm 1.51$ | $78.54 \pm 2.74$ | $78.15 \pm 1.32$ | $69.00 \pm 1.38$ | $50.80 \pm 1.15$ |
| PROTEINS | $72.05 \pm 5.77$ | $70.00 \pm 2.44$ | $68.47 \pm 5.00$ | $72.97 \pm 4.87$ | $57.67 \pm 2.62$ |
| BBBP | $85.90 \pm 0.99$ | $83.54 \pm 0.37$ | $84.78 \pm 2.57$ | $84.12 \pm 2.09$ | $22.80 \pm 0.86$ |

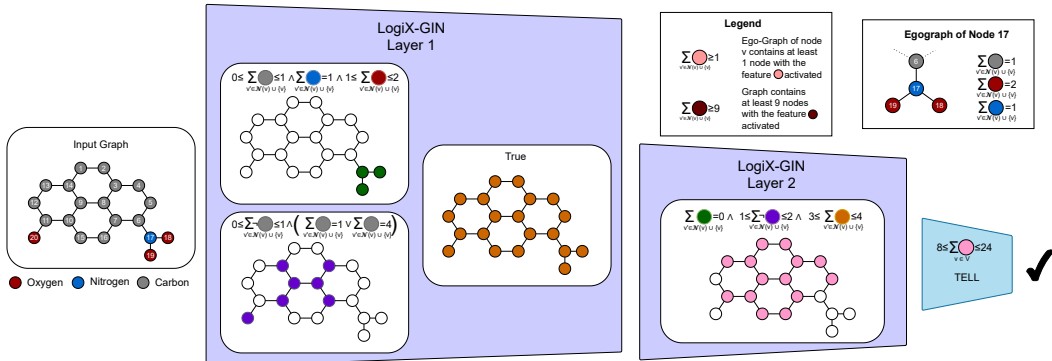

Figure 2: Layer-wise explanation of a LogiX-GIN model trained on the MUTAG dataset.

the $NO_2$ group, although this occurs within its inner layers. This illustrates that our pattern-based analysis, when limited to the last layer, may offer only a partial view of the model's reasoning. On the Mutagenicity dataset, LogiX-GIN identifies simpler patterns such as successive nitrogen atoms and the $NO_2$ group. GraphTrail similarly detects these patterns in the black-box model, while also uncovering additional carbon-related structures. On BA2Motifs, both identify the house motif, while on BAMultiShapes, only LogiX-GIN recovers the ground-truth rule.

Overall, although both methods highlight chemically and structural meaningful patterns, Logix-GNN exhibits less complexity and higher versatility than GraphTrail. More importantly, the key difference lies in the "alignment" between the model's behavior and the explanations. Alignment is defined as the accuracy between the model's predictions and the rule activations. GraphTrail, being post-hoc, may misalign with model behavior, while LogiX-GIN's explanations are fully aligned by design.

**Explaining the single layers of LogiX-GIN.**    In this section, we analyze the core functionality of our approach, which enables inspection of the model at each individual layer. In Figure 2, we present a real-world example showing the actual rules learned by LogiX-GIN on the MUTAG dataset. This example provides insight into how the model learns to predict molecular mutagenicity through interpretable rule-based reasoning.

As a first observation, although the architecture consists of 5 convolutional layers, only 2 are actively used in the decision process. Notably, the first layer captures low-level features, such as atom type and connectivity, which are then leveraged to identify functional groups. The first rule (green), in fact, detects the $NO_2$ group by counting the number of nitrogens, oxygens, and carbons. Interestingly, to enhance specificity, the model also checks on the number of carbons to detect the $NO_2$. The second rule (violet) is instead more complex, likely related to aromatic rings: the model counts the number of carbon atoms in each atom's neighborhood. Additionally, it also focuses on non-carbon atoms connected to only one carbon, which may help in identifying cyclic functional groups. The third feature (orange) activates across all nodes. While this might initially seem trivial, it serves an important role in the subsequent layer. In the second layer, in fact, features from the first layer are combined, and this feature is used for counting node connections. This offers valuable insight into the model's internal logic, showing how features that appear unrelated to the final prediction can still play a critical role.

Ultimately, the model's decision is determined by counting the number of nodes that activate a specific feature (pink) in the second convolutional layer. This mechanism sheds light on an important phenomenon: "removing graph parts may not affect predictions if key activations stay within the learned range". Consequently, even when a model appears to have lower fidelity w.r.t. node removal, this may be due not to reduced explainability, but to its reliance on counting-based decision logic.

**Analyzing rules' complexity.**    We now analyze how the models use the layers to learn logic rules. With this aim, we analyze the models after the pruning procedure and study how layers learn rules. Figure 3 reports the amount of rules learned by each layer of the models. For each dataset, we select only one seed and one class and plot the number of rules extracted by each layer. In the cases of multiple classes, we use class 0 as the target. From this analysis, we exclude rules that provide no

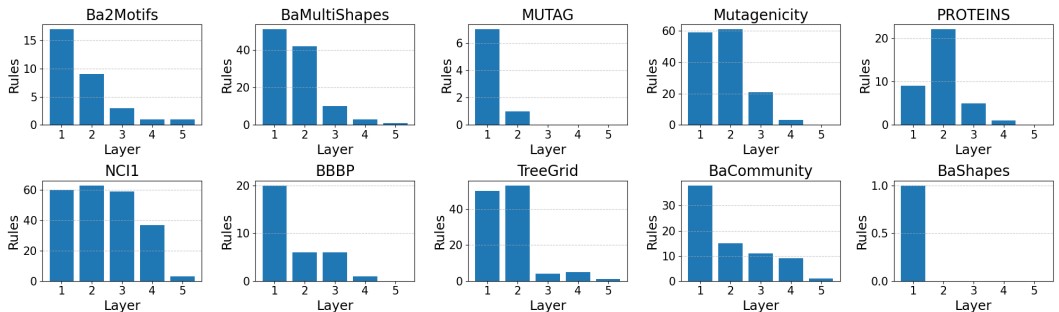

Figure 3: Amount of rules extracted by each convolutional layer on the 10 datasets in exam.

activations or activate on all the nodes, as they are mainly used to count node degrees. This analysis shows how models extract information from the graphs. Specifically, we observe that the first layers generally contain more rules, thus confirming the findings of the previous section. Indeed, the first layers tend to identify low-level patterns, which are then combined in the successive layers. We also see that in most cases the model does not use all the layers as the information extracted in the first layers is enough to perform the prediction. Contrarily, Mutagenicity and NCI1 are the datasets that require the highest amount of literals to describe the activity. This is most likely due to the higher complexity of the tasks and the high variability of the molecular graphs.

Overall, this analysis highlights the hierarchical nature of logic rules in the models. Early layers are densely populated with rules that extract fundamental features, while deeper layers serve a more selective role. This result is also confirmed when analysing rules activations (Appendix G). Furthermore, the redundancy of deeper layers in several datasets suggests a degree of architectural overcapacity, which pruning helps to address. Finally, the number of literals required across datasets underscores the varying complexity of the tasks and the richness of the input graphs, particularly in chemically diverse datasets. These interpretable insights directly inform model design: "no rules after layer 2 imply excess layers can be removed".

## 6   Conclusions

In this work, we introduced LogiX-GIN, a novel GNN architecture that incorporates logic-based reasoning directly into its layers to achieve self-explainability. We demonstrated that each layer of the architecture can be converted into logical formulas in GML formalism, offering a transparent and faithful representation of the model's reasoning. Through a series of experiments on both synthetic and real-world datasets for graph and node classification, we showed that LogiX-GIN maintains competitive performance while offering a new level of interpretability.

Despite these promising results, our work has some limitations. First, LogiX-GIN does not take into account edge features, limiting its potential expressiveness. Moreover, the usage of sigmoid activations and learnable binarization introduces training challenges, particularly in deeper networks or larger graphs. Although we mitigate these issues with a distillation-based pretraining strategy, the computational cost remains significant as the training is composed of multiple steps. Also, thresholding on input features might not be interpretable in cases where they are not attributed with specific semantic meanings. Future work could investigate case the integration with prototype-based or concept-based techniques that could improve interpretability thanks to their case-based reasoning. Another important limitation concerns the expressivity of the model itself: since LogiX-GIN relies on GML, its reasoning capabilities are constrained by what GML can express. As a result, the model may not be suitable for tasks that require more expressive or higher-order forms of reasoning beyond the scope of GML. Future research can address these limitations by exploring more scalable architectures and optimization techniques that preserve interpretability while improving computational efficiency. Additionally, extending the model to support a broader range of graph tasks or other logic formalisms might be interesting. In this case, combining logic-based reasoning with other forms of explanation, such as prototypes, may further enhance performance and interpretability.

## Acknowledgments and Disclosure of Funding

This work was supported by French state aid managed by the National Research Agency under the France 2030 program, with the references "WAIT4 ANR-22-PEAE-0008", "PANDORA ANR-24-CE23-0950", and "PORTRAIT ANR-22-CE23-0006".

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

# A Fourier Step Function

In this section, we analyze the functionality of the $\beta$ function of LogiX-GIN as defined in Equation 5. Figure 4 shows three examples illustrating how the parameters w (width) and b (bias or displacement) affect the shape and position of the intervals. The first case ($w = 1, b = 0$) shows a baseline case where, in this interval of input values ($-5 < x < 5$), the output corresponds to a 1 if $x < -3$ or $0 < x < 3$. The second case ($w = 2, b = 0$) highlights how $w$ can be used to modify the width of the intervals. Finally, the third case ($w = 1, b = -2$) shows how the parameter $b$ is instead responsible to apply a displacement of the thresholds (in this case of $-2$).

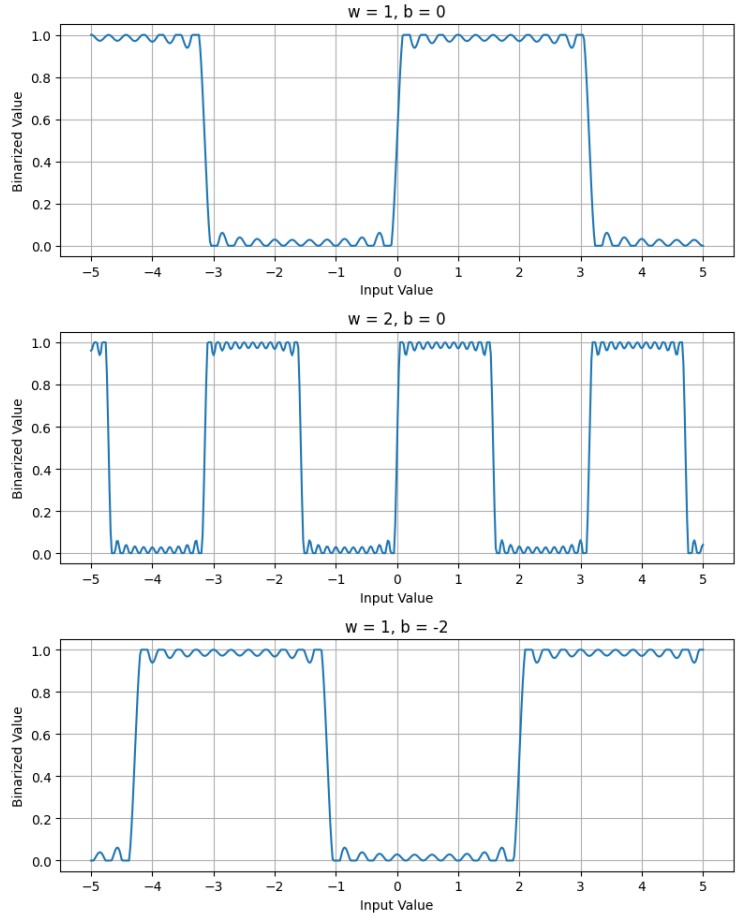

Figure 4: Examples of the binarization using the Fourier step function in Equation 5

# B Examples of Graded Modal Logic Formulas

Following Barceló et al. [4], we use colors as unary predicates: *Blue(x)*, *Red(x)*, and *Green(x)* denote that node $x$ is respectively blue, red, or green.

**Example 1: Atomic predicate.**
$$\varphi(x) := Blue(x)$$
This formula holds if $x$ is blue.

**Example 2: At least two red neighbors.**
$$\varphi(x) := \exists^{\geq 2} y \left( E(x, y) \wedge Red(y) \right)$$
This states that $x$ has at least two neighbors that are red.

**Example 3: Blue node without red neighbors.**

$$\varphi(x) := Blue(x) \wedge \neg\exists^{\geq 1}y\left(E(x,y) \wedge Red(y)\right)$$

This expresses that $x$ is blue and has no red neighbors.

**Example 4: Disjunction of neighborhood conditions.**

$$\varphi(x) := \left(\exists^{\geq 3}y\left(E(x,y) \wedge Red(y)\right)\right) \vee \left(\exists^{\geq 1}y\left(E(x,y) \wedge Green(y)\right)\right)$$

This means that $x$ has either at least three red neighbors, or at least one green neighbor.

**Example 5: Nested neighborhood property.**

$$\varphi(x) := \exists^{\geq 1}y\left(E(x,y) \wedge \exists^{\geq 1}z\left(E(y,z) \wedge Green(z)\right)\right)$$

This states that $x$ has a neighbor $y$ which itself has at least one green neighbor $z$.

These examples illustrate how GML formulas, written with colored predicates, capture local structural patterns in graphs using counting quantifiers and logical connectives.

## C   LogiX-GIN Explanation Procedure

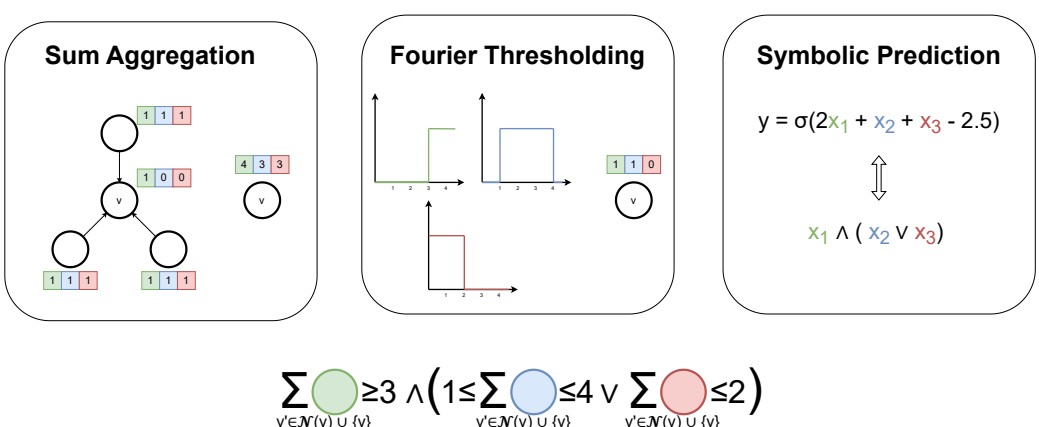

Figure 5: Diagram of LogiX-GIN layer information flow and explanation procedure. At the first step, the aggregated node features of node $v$ are obtained by summing up over its neighbors. Successively, Fourier thresholding allows to activate only when features lie in specific intervals by generating a binary representation. Finally, a TELL-like classifier outputs a value whose activation can be directly converted into a DNF formula. Finally an overall rule is obtained by merging the thresholds into the DNF extracted by the last step. In The final rule, the output feature of for node $v$ will be one if and only if the neighboorhod contains at least 3 nodes with the green feature activated and either there are at most two nodes with the red feature activated or between 1 and 4 nodes with the blue features activated.

**Overview and pipeline.**   Figure 5 shows the LogiX-GIN pipeline. Node features from a target node and its neighborhood are first aggregated by summation. These aggregated values are then binarized by a learnable Fourier step function, which converts counts into discrete on/off literals. The binarized signals are passed through a logic layer with non-negative weights, producing interpretable logical activations. During training, a teacher GIN model is used for distillation, and Hoyer-style pruning can be applied to remove redundant rules. From this pipeline, we can extract explanations at different levels of granularity:

- **Layer-by-layer explanations** provide a fine-grained analysis representing the exact model behavior. Each hidden layer corresponds to a set of logic rules that are directly readable from the parameters of its logic transformation. For every output unit, the threshold conditions

on its binary inputs define minimal logical clauses. Presenting rules per layer allows us to reconstruct how evidence is accumulated step by step, without needing to collapse the entire network into a single formula.

- **Global logic rules** identify logic formulas over subgraphs that globally approximate the model's behavior. To obtain them, at the final classifier, we extract global rules that summarize the decision process for the whole graph. The extraction procedure follows the following step:

  1. Apply rule extraction to the final logic layer, obtaining global rules associated with each class.
  2. Identify the nodes in a given graph that activate these rules.
  3. Collect the induced neighborhoods of these nodes and compute canonical subgraphs.
  4. Use a standard graph isomorphism algorithm [7] to merge isomorphic subgraphs and identify recurring motifs.

  The resulting set of global rules provides human-readable summaries such as graphs containing motif $M$ are classified as class $c$". To evaluate these rules, we measure their alignment, that is, the agreement between the rule activations and the model's predictions on held-out data.

- **Node attributions from global rules** provide scores for nodes of a graph which measure their contribution towards a prediction. When a global rule is activated for a graph, we can attribute responsibility to the specific nodes that support its conditions. A node is considered relevant if it appears in at least one minimal conjunct that satisfies the rule. In this way, the explanation highlights the exact part of the graph that triggered the decision. This yields node-level attributions that are directly derived from the logic rules, avoiding the need for post-hoc gradient or perturbation methods.

## D  Demonstration

**Proposition 1.** *Let $h_v^{(k)}$ denote the output of the $k$-th layer of the LogiX-GIN architecture, defined as in Equation 4, where $\lambda^{(k)}$ is a monotonic logic transformation as in Equation 7, and $\beta^{(k)}$ is a differentiable binarization function over learned activation intervals as in Equation 5. Then, for each $k$, there exists a set of formulas $\{\varphi_j^{(k)}(x)\}_j$ in the syntax of GML such that*

$$h_v^{(k)}[j] = 1 \iff (G, v) \models \varphi_j^{(k)}(x),$$

*where $G$ is the input graph and $x$ is the free variable corresponding to node $v$.*

*Proof.* We proceed by structural correspondence between the computational components of $h_v^{(k)}$ and the semantic constructs of GML:

- **Aggregation.** Given a node $v \in V$, the expression

$$a_v := \sum_{u \in \mathcal{N}(v) \cup \{v\}} h_u^{(k-1)} \tag{8}$$

  computes, for each feature index $i \in \{1, ..., d\}$, with $d$ the hidden dimension, the number of neighbors $u$ such that $h_u^{(k-1)}[i] = 1$. This results in a count vector $a_v \in \mathbb{N}^d$, where each entry encodes the multiplicity of a property among the neighbors.

- **Binarization.** The function $\beta^{(k)}$ applies learnable thresholding to each coordinate $a_v[i]$, producing:

$$\tilde{a}_v^{(k)} := \beta^{(k)}(a_v) \in \{0, 1\}^d. \tag{9}$$

  Each output $\tilde{a}_v^{(k)}[i] = 1$ encodes the satisfaction of a threshold condition:

$$\tilde{a}_v^{(k)}[i] = 1 \iff a_v[i] \in \Theta_i \iff |\{u \in \mathcal{N}(v) \cup \{v\} : h_u^{(k-1)}[i] = 1\}| \in \mathcal{I}_{\beta^{(k)}, i}, \tag{10}$$

where $\mathcal{I}_{\beta^{(k)},i}$ is a learnable interval determined by $\beta^{(k)}$ for the $i$-th dimension, as defined in Equation 6. By inductive hypothesis, for each $i$, there exists a formula $\psi_i^{(k-1)}(x)$ such that $h_u^{(k-1)}[i] = 1 \iff (G, u) \models \psi_i^{(k-1)}(x)$. Therefore:

$$a_v^{(k)}[i] = 1 \iff (G, v) \models \bigvee_{(t_i, T_i) \in \mathcal{I}_{\beta^{(k)}}} \left( \left( \exists^{\geq t_i} y \left( E(x, y) \wedge \psi_i^{(k-1)}(y) \right) \right) \right.$$
$$\left. \wedge \left( \exists^{\leq T_i} y \left( E(x, y) \wedge \psi_i^{(k-1)}(y) \right) \right) \right). \tag{11}$$

Hence, each $a_v^{(k)}[i]$ corresponds to a valid graded modality in GML.

- **Monotonic Logic Layer.** As shown by Ragno et al. [23], due to non-negativity of weights and the monotonicity of $\sigma$, each output neuron $h_v^{(k)}[j]$ from Equation 7 corresponds to a logic formula in disjunctive normal form (DNF):

$$h_v^{(k)}[j] = 1 \iff \bigvee_{S \in \mathcal{S}_j} \bigwedge_{i \in S} \tilde{a}_v^{(k)}[i] = 1, \tag{12}$$

wher $\mathcal{S}_j$ is the set of subsets of features with corresponding weights that sum up to a value greater than $-b_j^{(k)}$. Since each $a_v^{(k)}[i]$ corresponds to a GML formula as shown above, it follows that:

$$h_v^{(k)}[j] = 1 \iff (G, v) \models \varphi_j^{(k)}(x),$$

where

$$\varphi_j^{(k)}(x) := \bigvee_{S \in \mathcal{S}_j} \bigwedge_{i \in S} \bigvee_{(t_i, T_i) \in \mathcal{I}_{\beta^{(k)}}} \left( \left( \exists^{\geq t_i} y \left( E(x, y) \wedge \psi_i^{(k-1)}(y) \right) \right) \right.$$
$$\left. \wedge \left( \exists^{\leq T_i} y \left( E(x, y) \wedge \psi_i^{(k-1)}(y) \right) \right) \right). \tag{13}$$

By induction on $k$, and given that the base case (input binarization) is definable by unary predicates, we conclude that all $h_v^{(k)}[j]$ are convertible into GML formulas. Thus, the output of the LogiX-GIN architecture is fully convertible in GML. $\qquad \square$

# E   Node Attribution Evaluation

Here, we perform an analysis to evaluate the explanations proposed by different approaches when simply focus on node attributions. We perform our comparison with 5 post-hoc methods (GNNExplainer [31], PGExplainer [15], Integrated Gradients [27], SubgraphX [34], and GStarX [35]) applied on the black-box GIN model and 2 self-interpretable models (PiGNN [22] and GIB [32]) We perform the evaluation in terms of the Fidelity of the attributions. Fidelity is defined as the ratio of graphs whose prediction is shifted when removing the most important nodes.

In literature, node attribution evaluation is mainly designed of post-hoc instance-level approaches, which focus on finding the most important nodes that lead to a specific prediction. On the contrary, self-interpretable models are designed to provide visual explanations thanks to architectural constraints. For these reasons, node attributions represent only an approximation of the explanatory power of self-interpretable models and this evaluation setting can end up being unfair to them. This is primarily due to a lack of evaluation methods for self-interpretable models in literature for several reasons. First, these type of models are still highly unexplored, second, there is often a high diversity between the various approaches. For instance, prototype-based approaches use the similarity between prototypes and subgraphs of the input graph to determine the class. Node attributions for these methods are obtained by extracting such similarities. However, it is not guaranteed that removing the portions of the input graphs that actually led to a specific prediction cause a shift in the prediction. On the contrary, optimization-based methods such as GStarX that specifically aims to find subgraphs that when removed provoke shifts in the prediction. Additionally, while with post-hoc methods are generally compared when applied on the same model, in the case of self-explainable models, we have different models under analysis.

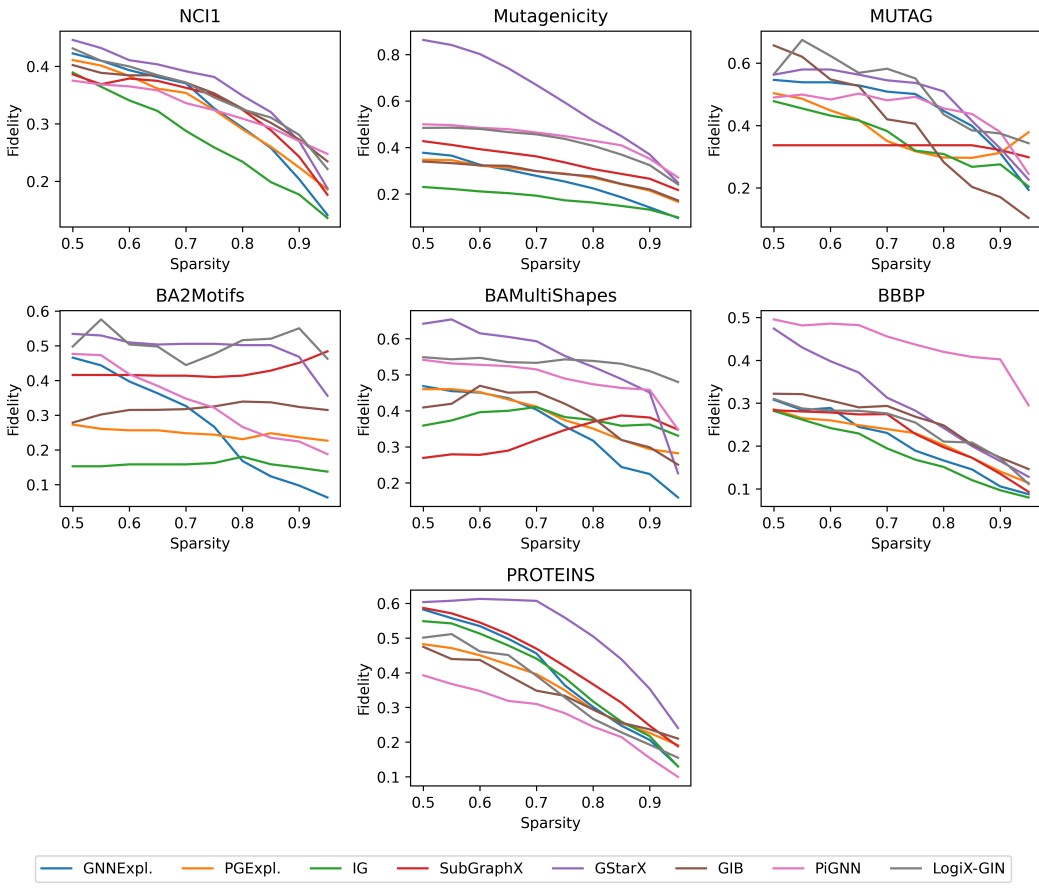

Figure 6: Fidelity.

Figure 6 shows the Fidelity score on the graph classification datasets. Since the different methods report continuous attribution score, the fidelity is calculated on the hard masks at different sparsity level. In particular, we vary the sparsity between 0.5 and 0.95.

We observe that on four cases, namely NCI1, MUTAG, BA2Motifs and BAMultiShapes, LogiX-GIN produces Fidelity scores that are comparable or better that the state-of-the-art post-hoc approaches, specifically with optimization-based ones. On BBBP, Mutagenicity and PROTEINS, instead we observe better values from GStarX and PiGNN. However, as already underlined, in the case of LogiX-GIN, explanations are obtained by simply looking at the rules activated on the last layer, which might not truly indicate its true behavior. For this reason, we believe that a more thorough examination of the model through inspection of its complete logic rules can actually provide better transparency of the model.

# F   Last Layer Rule Activations

This section reports the activations of the logic rules of the last layers on the datasets of the global rules experiment. Figure 7 shows graphs of the validation set highlighting the nodes that activate the rules of the last layer. In order to provide a compact representations, activations are aggregated using isomorphism. Therefore, for each dataset, only non-isomorphic activations are shown. We observe that the rules capture patterns that are generally known to be related with the predicted class.

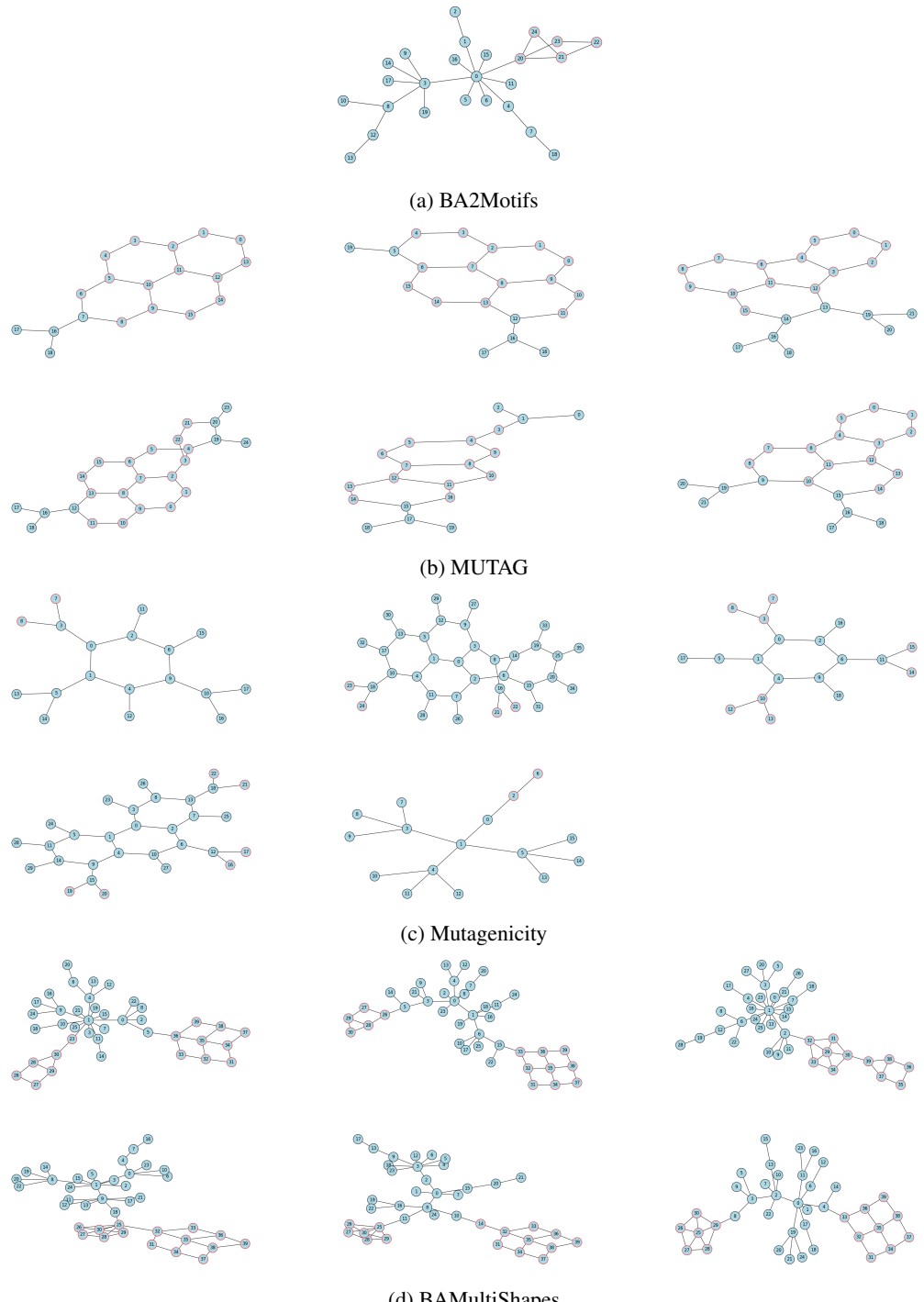

(a) BA2Motifs

(b) MUTAG

(c) Mutagenicity

(d) BAMultiShapes

Figure 7: Last layer rules activations for obtaining global explanations. For each dataset, we show validation set graphs that activate for the rules and highlight in red the nodes corresponding to such activations.

# G   Logic Rules Activations

This section reports the activations of the logic rules on the 10 datasets. For each model we extract the rules of each layer and we plot in Figure 8 the distribution on the percentage of the nodes that are

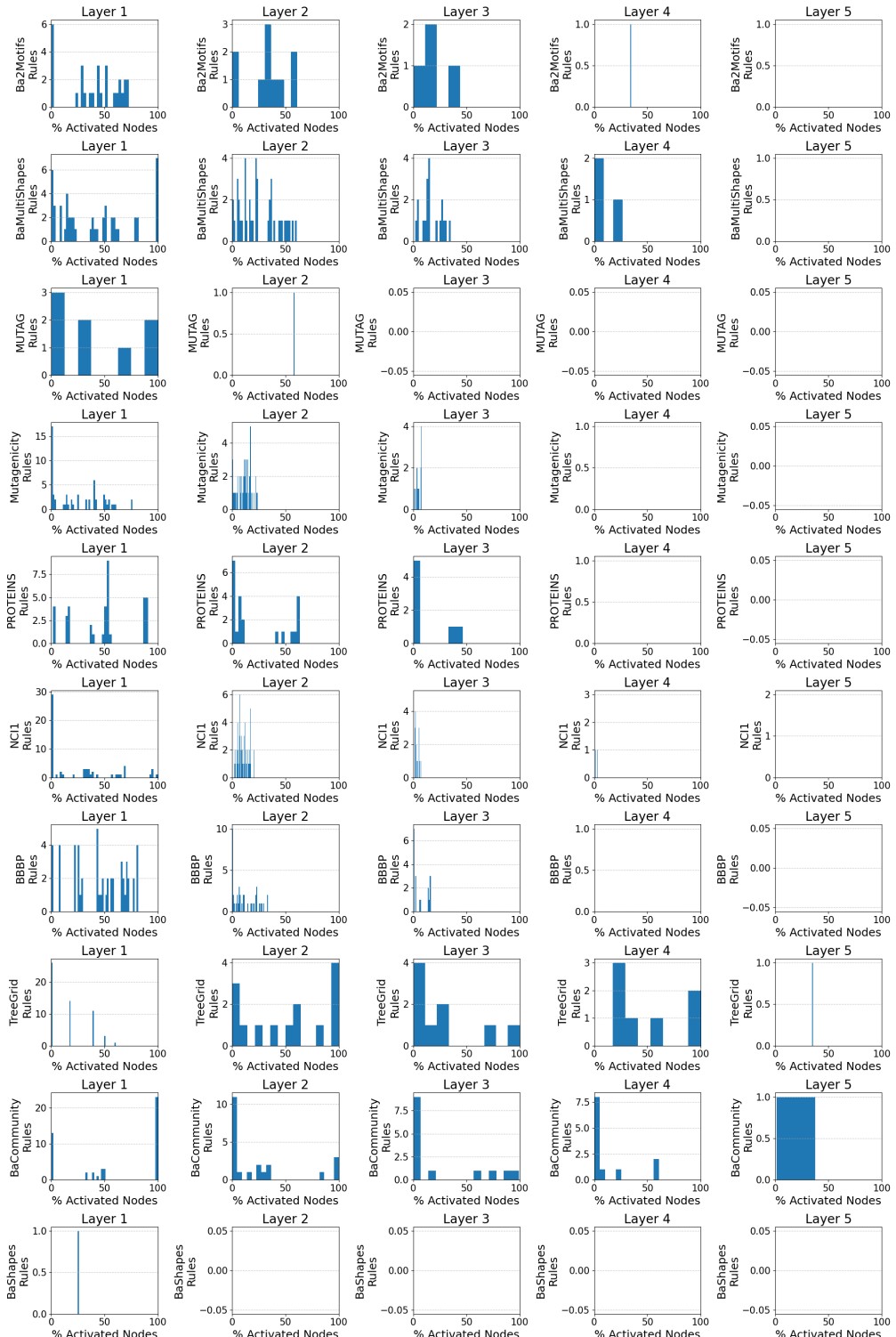

Figure 8: Analysis on percentage rules activations for each layer in LogiX-GIN.

activated. We observe as a general pattern that early layers have high variations on the activations, most likely due to the fact that they tend to recognise more general patterns. On the contrary later layers focus on more detailed patterns, drastically reducing the percentage of nodes that are activated.

# H   Ablation Study on Pruning

We perform an ablation study on the BBBP and BA2MOTIFS datasets to evaluate the effect of pruning strategies. We compare (i) *layer-by-layer pruning*, where the Hoyer regularization is applied sequentially starting from the last layer, and (ii) *full pruning*, where it is applied jointly to all layers. Table 3 reports the percentage decrease of non-zero weights (from the unpruned to the pruned model) for both strategies.

Table 3: Ablation study on pruning. We report the percentage decrease of non-zero weights after pruning. Layer-by-layer pruning consistently yields stronger reductions compared to full pruning.

| Dataset | Layer-by-layer | Full |
|---------|----------------|------|
| Ba2Motifs | 83.13 | 78.84 |
| BBBP | 30.39 | 0.00 |

The results show that the layer-by-layer strategy better reduces the number of non-zero weights. In contrast, jointly optimizing all layers at once leads to weaker pruning due to interdependencies among rules: if a rule at layer $\ell$ depends on features from layer $\ell - 1$, then pruning must respect this order. For this reason, pruning all layers together may incorrectly remove weights that are still needed in subsequent rules.

# I   Hyperparameters, Resources and Reproducibility

All the experiments, including those on GraphTrail, post-hoc methods and other self-explainable architectures, are documented in the supplementary material. The experiments were performed on a machine equipped with an Intel(R) Core(TM) i9-10900K CPU @ 3.70GHz and an NVIDIA GeForce RTX 4090. The implementation is done in PyTorch and PyTorch-Geometric and all the library versions are detailed in the environment specifics available in the supplementary material. In Table 4 and Table 5 we detail the hyperparameters of the models.

In Table 6 we report the average training times of GIN and LogiX-GIN. Although the LogiX-GIN training times are much higher compared to the ones of the black-box, it is always important to keep in mind that they allow for a full inspection of the single layers of the model. Before LogiX-GIN, no model or technique allowed for such level of explainability. Additionally, it is also important to keep in mind that, in the case of the GIN model, in order to have an explanation only of the last layer, we would still need to execute GraphTrail, which might take days to generate an explanation.

Table 4: GIN Models Hyperparameters

| Dataset | batch_size | dropout | epochs | hidden_dim | l2 | lr |
|---------|-----------|---------|--------|-----------|------|------|
| BaShapes | 32 | 0 | 3000 | 32 | 0.0001 | 0.001 |
| BaMultiShapes | 128 | 0.5 | 3000 | 32 | 0.0001 | 0.001 |
| BaCommunity | 128 | 0.5 | 3000 | 16 | 0.0001 | 0.01 |
| Ba2Motifs | 128 | 0 | 3000 | 16 | 0.0001 | 0.001 |
| BBBP | 32 | 0.5 | 3000 | 16 | 0.0001 | 0.001 |
| MUTAG | 128 | 0.5 | 3000 | 32 | 0.0001 | 0.001 |
| NCI1 | 128 | 0 | 3000 | 32 | 0.0001 | 0.001 |
| PROTEINS | 32 | 0 | 3000 | 32 | 0.0001 | 0.001 |
| Mutagenicity | 32 | 0.5 | 3000 | 32 | 0.0001 | 0.001 |
| TreeGrid | 128 | 0 | 3000 | 32 | 0.0001 | 0.001 |

# J   Rule Pruning Statistics

Here, we report an analysis of the effects of our pruning strategy on the LogiX-GIN model. Table 7 shows the test accuracy and the number of non-zero weights before and after pruning on all the datasets in exam. We observe that the strategy does not impact the performances drastically. On the

Table 5: LogiX-GIN Models Hyperparameters.

| Dataset | batch_size | conv_reg | epochs | fc_reg | l2 | lr |
|---|---|---|---|---|---|---|
| BaShapes | 128 | 0.001 | 5000 | 0.01 | 0 | 0.01 |
| BaMultiShapes | 32 | 0.001 | 5000 | 0.01 | 0.0 | 0.001 |
| BaCommunity | 32 | 0.001 | 5000 | 0.1 | 0.0001 | 0.01 |
| Ba2Motifs | 32 | 0.001 | 5000 | 0.01 | 0.0 | 0.001 |
| BBBP | 32 | 0.001 | 5000 | 0.01 | 0.0 | 0.001 |
| MUTAG | 32 | 0.001 | 5000 | 0.01 | 0.0 | 0.01 |
| NCI1 | 32 | 0.001 | 5000 | 0.1 | 0.0001 | 0.0001 |
| PROTEINS | 32 | 0.001 | 5000 | 0.01 | 0.0 | 0.001 |
| Mutagenicity | 128 | 0.001 | 5000 | 0.01 | 0.0001 | 0.001 |
| TreeGrid | 32 | 0.001 | 5000 | 0.01 | 0.0 | 0.001 |

Table 6: Average and standard deviation of training times (in milliseconds) for GIN and LogiX-GIN across datasets (updated values).

| Dataset | GIN (avg ± std) | LogiX-GIN (avg ± std) |
|---|---|---|
| Ba2Motifs | 2727.00 ± 207.80 | 6671.67 ± 177.83 |
| BaMultiShapes | 1193.67 ± 35.37 | 6354.00 ± 201.92 |
| MUTAG | 279.00 ± 122.06 | 797.00 ± 26.87 |
| Mutagenicity | 3656.00 ± 40.90 | 5748.33 ± 27.38 |
| PROTEINS | 1872.33 ± 13.89 | 11077.00 ± 4.97 |
| NCI1 | 1695.33 ± 48.80 | 49646.67 ± 636.76 |
| BBBP | 2044.00 ± 77.94 | 19855.00 ± 46.50 |
| TreeGrid | 265.33 ± 19.94 | 4361.00 ± 12.03 |
| BaCommunity | 497.33 ± 11.09 | 21057.67 ± 3.77 |
| BaShapes | 1072.67 ± 73.19 | 16680.67 ± 4.50 |

Table 7: Statistics of classification performances and number of non-zero weights on the LogiX-GIN model before and after pruning. The experiments are performed over 5 seeds and mean and standard deviation are reported for statistical significance.

| Dataset | Stage | Accuracy | Total Non-zeros | Conv-Layer 1 | Conv-Layer 2 | Conv-Layer 3 | Conv-Layer 4 | Conv-Layer 5 | FC Layer |
|---|---|---|---|---|---|---|---|---|---|
| Ba2Motifs | Original | 1.00 ± 0.00 | 4136 ± 432 | 252 ± 48 | 937 ± 112 | 958 ± 56 | 843 ± 51 | 1007 ± 127 | 137 ± 70 |
| | Pruned | 1.00 ± 0.00 | 880 ± 581 | 31 ± 16 | 201 ± 211 | 334 ± 380 | 67 ± 72 | 243 ± 399 | 2 ± 0 |
| BaMultiShapes | Original | 1.00 ± 0.00 | 4546 ± 119 | 280 ± 12 | 1090 ± 57 | 1013 ± 25 | 908 ± 53 | 1071 ± 81 | 182 ± 42 |
| | Pruned | 0.99 ± 0.01 | 3355 ± 265 | 196 ± 34 | 959 ± 134 | 782 ± 264 | 550 ± 434 | 829 ± 360 | 37 ± 54 |
| MUTAG | Original | 0.87 ± 0.04 | 2295 ± 362 | 228 ± 21 | 660 ± 86 | 444 ± 158 | 498 ± 121 | 389 ± 141 | 73 ± 20 |
| | Pruned | 0.85 ± 0.04 | 711 ± 437 | 133 ± 85 | 236 ± 160 | 186 ± 218 | 107 ± 139 | 8 ± 11 | 40 ± 14 |
| Mutagenicity | Original | 0.82 ± 0.02 | 4643 ± 280 | 474 ± 26 | 987 ± 41 | 1000 ± 30 | 999 ± 111 | 976 ± 184 | 205 ± 15 |
| | Pruned | 0.81 ± 0.02 | 4133 ± 545 | 474 ± 26 | 987 ± 41 | 1000 ± 30 | 957 ± 161 | 651 ± 530 | 61 ± 74 |
| NCI1 | Original | 0.80 ± 0.02 | 4443 ± 139 | 495 ± 32 | 827 ± 46 | 942 ± 102 | 952 ± 65 | 1061 ± 68 | 165 ± 9 |
| | Pruned | 0.80 ± 0.02 | 4197 ± 304 | 495 ± 32 | 827 ± 46 | 942 ± 102 | 952 ± 65 | 956 ± 208 | 23 ± 2 |
| PROTEINS | Original | 0.72 ± 0.06 | 2547 ± 509 | 143 ± 10 | 796 ± 163 | 512 ± 175 | 568 ± 256 | 446 ± 380 | 80 ± 10 |
| | Pruned | 0.71 ± 0.03 | 1346 ± 468 | 107 ± 41 | 739 ± 155 | 190 ± 224 | 230 ± 263 | 1 ± 2 | 77 ± 5 |
| BBBP | Original | 0.88 ± 0.01 | 2775 ± 510 | 315 ± 11 | 913 ± 68 | 687 ± 194 | 587 ± 294 | 155 ± 303 | 116 ± 20 |
| | Pruned | 0.86 ± 0.02 | 1732 ± 250 | 272 ± 91 | 740 ± 373 | 481 ± 327 | 154 ± 300 | 0 ± 0 | 83 ± 57 |
| TreeGrid | Original | 0.98 ± 0.01 | 4014 ± 82 | 189 ± 15 | 860 ± 46 | 965 ± 74 | 970 ± 54 | 1008 ± 52 | 20 ± 14 |
| | Pruned | 1.00 ± 0.01 | 1007 ± 404 | 99 ± 19 | 333 ± 107 | 317 ± 243 | 205 ± 101 | 46 ± 35 | 5 ± 2 |
| BaShapes | Original | 0.93 ± 0.03 | 1016 ± 104 | 68 ± 9 | 241 ± 47 | 220 ± 29 | 220 ± 22 | 195 ± 27 | 69 ± 24 |
| | Pruned | 0.97 ± 0.01 | 482 ± 124 | 58 ± 10 | 164 ± 50 | 133 ± 67 | 51 ± 7 | 26 ± 5 | 48 ± 36 |
| BaCommunity | Original | 0.86 ± 0.02 | 3621 ± 272 | 193 ± 18 | 739 ± 57 | 706 ± 110 | 688 ± 100 | 767 ± 76 | 525 ± 77 |
| | Pruned | 0.89 ± 0.02 | 1248 ± 341 | 118 ± 37 | 384 ± 177 | 297 ± 194 | 170 ± 65 | 99 ± 27 | 179 ± 183 |

contrary, in some cases we also record an improvement of the performances as the pruning encourages generalization in the model. The amount of non-zero weights reflects the number of rules identified in the experimental section. Indeed, we observe that the number of non-zero weights is directly linked with the number of rules that the model uses.

