# OpenReview forum: "On Logic-based Self-Explainable Graph Neural Networks"
_NeurIPS.cc/2025/Conference — NeurIPS 2025 poster_

### Official Review · Reviewer_Qw5B · 2025-06-21

**Clarity:** 3
**Significance:** 2
**Originality:** 1
**Rating:** 4
**Confidence:** 5

**Summary:**

The paper presents an interpretable graph neural network. Its interpretability stems from binarizing each layer’s input and constraining the layer weights to be non-negative. These weights are translated into graded modal logic expressions that are more suitable for GNNs than binary expressions. The pipeline goes like this: Node features are binarized using TELL’s preprocessing step, which learns weights to threshold the features. Aggregating the binary features generates an integer-valued vector. A parametric Fourier step function is used to binarize it. The step function’s weights learn the integer ranges in which dimensions activate, acting as a proxy for counting. The ability to count facilitates the use of graded modal logic. The authors mention that their use of the sigmoid function hinders initial training and suggest pre-training the interpretable layers against a black-box GIN, where the interpretable layers emulate the GIN layers’ weights binarized using Gumbel-sigmoid activations. The proposed method’s novelty lies in using the Fourier step function as a proxy for counting facilitating the use of TELL layers on graphs.

**Questions:**

1. How is the non-negative constraint on weights preserved during gradient descent? Methods like non-negative matrix factorization (NMF) use hand-crafted updates to ensure such behavior.

2. I have assumed the input node features are binarized using TELL’s preprocessing as mentioned in line 168. Is this correct?

3. Line 223: Is the output of the sigmoid function binarized? If yes, what threshold is used?

4. Why are the layers pruned individually, and why do it in reverse?

5. How is the black-box GIN modified for pre-training so that the interpretable layer can mimic it? For example, are the hidden weights binarized actively during training or at the end of each epoch? Please provide these details.

6. Is “alignment” the rule’s accuracy against the model’s predictions? And in LogiX-GIN, the model and rules are one and the same resulting in 100\% alignment. Is my understanding correct?

7. Please provide a figure similar to figure 2 for the non-mutagenic class of MUTAG and the negative class in BA2Motifs. I’m curious to see the explanations of the negative classes.

8. Line 311: “exhibits less complexity and higher versatility than GraphTrail” What does versatile mean here? I contest the complexity part. Table 2 shows 651 active rules for Mutagenicity at layer 5 compared to 1 rule given by GraphTrail.

9. Given the 243 active rules for BA2Motifs, how did the authors single out the explanation shown in figure 1. Same question for 829 active rules for BAMultiShapes, and Mutagenicity with 651 rules.

**Ethical Concerns:**

["NO or VERY MINOR ethics concerns only"]

**Final Justification:**

In light of my discussion with the authors, I am raising my score from **reject** to a **borderline accept**.

I still find the novelty limited by NeurIPS standards. However, there is precedent for similar extensions. For instance, one of the works cited by the authors, LEN [4], has been employed similarly for GNN explanations in GLGExplainer (ICLR 2023).

I found the experimental section in the paper unconvincing. However, the additional experiments, ablations, and clarifications provided in the rebuttal have positively swayed my assessment.

That said, I still find the alignment experiment quite odd and misleading, and I don’t think it adds value to the paper.

I appreciate the authors’ acknowledgment of certain limitations, which helps clarify the scope of the work.

Above all, the paper most certainly needs a detailed methodology subsection which demonstrates extraction of explanations in the form of graded modal logic, as that is THE selling point of the paper. While the authors have indicated that this is implemented in code, it is essential that it be presented explicitly in the paper.

In its current form, which parts of the pipeline are intended for interpretability and how remains a bit ambiguous to me. There are multiple potential candidates and their interplay: weights, GML equivalents, activations, and isomorphic subgraphs. The paper would benefit from describing the intended use of each these aspects of the model’s output from an end-user's perspective. The authors have partially addressed this in the rebuttal, and incorporating that clarification into the paper would aid future readers.

**Limitations:**

The authors have mentioned most of their limitations. There are two that should also be included:
1. The inability to utilize edge features should be stated in the limitations.

2. Input binarization is appropriate when dimensions are independent and semantically meaningful—such as one-hot encoded atom types, categorical indicators, or continuous scalar features with clear interpretations (e.g., atomic radius). However, it is not suitable for dense embeddings like GloVe, where individual dimensions lack interpretability and meaning is distributed across the entire vector. Thresholding assumes each component is interpretable, which is not the case with such embeddings. The presence of specific values in certain dimensions may not correspond to anything meaningful, as the semantics are not localized. This limits the datasets on which the proposed method remains interpretable and should be clearly stated as a limitation.

**Paper Formatting Concerns:**

Nil

**Quality:**

2

**Strengths And Weaknesses:**

# Strengths
1. Graded Modal Logic is a good fit for GNNs and is likely to motivate future research on global explanations.
2. The Fourier Step function is a neat way to binarize integer inputs.
3. The interpretable model matches black-box performance on molecular and synthetic data.
4. The authors have highlighted some of their model’s limitations.

# Weaknesses
1. **Limited originality:** The proposed method draws heavily from TELL (ECAI 24), with much of the architecture and pipeline inherited directly. The authors' contribution lies in using the Fourier step function as a proxy for counting facilitating the use of TELL layers on graphs. Other core aspects of the pipeline like input binarization, constrained positive weights and their conversion to graded modal logic are reused from TELL.

2. **Missing baseline**: GNAN (NeurIPS '24) is a natural baseline for this work, yet it has not been compared against. Here’s the citation for your reference: Bechler-Speicher, Maya, Amir Globerson, and Ran Gilad-Bachrach. "The intelligible and effective graph neural additive network." Advances in Neural Information Processing Systems 37 (2024): 90552-90578.

3. Line 118: “demonstrated to not be properly faithful” This statement needs supporting evidence.

4. **Binarizing vectors holding a semantic meaning**: Input binarization is appropriate when dimensions are independent and semantically meaningful—such as one-hot encoded atom types, categorical indicators, or continuous scalar features with clear interpretations (e.g., atomic radius). However, it is not suitable for dense embeddings like GloVe, where individual dimensions lack interpretability and meaning is distributed across the entire vector. Thresholding assumes each component is interpretable, which is not the case with such embeddings. The presence of specific values in certain dimensions may not correspond to anything meaningful, as the semantics are not localized. This limits the datasets on which the proposed method remains interpretable and should be clearly stated as a limitation.

5. **Sigmoid struggling on 5 layers**: The paper mentions that the model struggles with initial learning due to its use of sigmoid functions. GNN’s are typically 2-5 layers deep. The authors have also used 5 layers. That’s not deep. Depth isn’t the reason the model is struggling. An empirical proof needs to show that the detrimental effect on training at such a shallow depth is caused by the sigmoid function.

6. **Invalid pruning assumption**: Pruning layers individually seems problematic. Suppose we prune layer $L$, selecting a few prominent weights based on its current input. If we then prune layer $L−1$, its output changes, thus altering the input to layer $L$. This is problematic because layer $L$ was pruned based on the original input, which is no longer valid.

7. **Dataset diversity**: All the datasets used in the paper are synthetic or molecular. Experiments on continuous valued features, especially the ones with dense features, e.g., GloVe embeddings, are missing.  A case study similar to MUTAG is needed on such datasets.

8. **Spurious agreement between explanation and performance**: BAMultiShapes explanation is incorrect even though it exactly matches the one reported in GLGExplainer and PyTorch Geometric. It doesn't exclude all three motifs appearing together. Let me explain as there’s a bigger problem at hand. GLGExplainer’s authors have used one set of formulae to create the dataset (elaborated in their section 4.1) but visualized an incorrect version which has been picked up by PyG. Please see this [anonymous repo](https://anonymous.4open.science/r/bamultishapes_proof-BFD9) for the experiment proof. The formula presented in figure 1 and the corresponding results in table 1 makes me skeptical of the validity of the other results as well. That formula should not achieve 100% accuracy as the experiment should make clear.

9. **Too many rules to be considered interpretable**: Even in simple, synthetic datasets like BA2Motifs, BAShapes, and TreeGrids, the last layer has 243, 46 and 26 active rules post-pruning. In these datasets, the class is identified by a single motif. Why do we need 243 rules for it? This number goes to 651 and 956 for Mutagenicity and NCI1. These many rules, even though individually interpretable, cannot be considered human interpretable in any meaningful sense. The cognitive load of understanding all 500+ rules together is too high for a human to grasp. In contrast, post-hoc global explainers generate a single concise formula for each of these datasets offering significantly greater human interpretability.

10. **Instability:** The model is unstable. Table 2: BAMultiShapes has 829 active rules with an std of 360; Mutagencity has 651 active rules with an std of 530.

11. **Ablations:** The authors should conduct the following ablation studies:
    - Validate the need for pre-training by presenting results with and without pre-training.
    - Pruning layers individually vs. pruning the model as a whole.

12. The inability to utilize edge features should be stated in the limitations.

13. Figure 6 is poorly visualized: nodes are very small; they should be colored instead of their outline being highlighted (which is barely visible); node IDs are irrelevant to the figure and should be replaced by node types.

14. Minor corrections:
    - Line 140: “using from” a word seems to be missing.
    - Figure 2’s font is too small.
    - In figure 10, $v’ \in N(v)$ should be replaced with $v’ \in N(v) \cup {v}$
    - Include the number of available cores in section G.
    - Start the checklist on a new page.

---

> ### Author Rebuttal · Authors · 2025-07-27
>
> We thank the reviewer for the suggestions and comments.
> 1. **Limited originality.** We understand the concerns of the reviewer. However, we would like to point out that integrating such constraints in the graph domain is not trivial and has required several changes in the architecture. In TELL, all the experiments were conducted with 1-layer model and no pruning strategies were presented.
> 2. **Missing baseline.** We thank the reviewer for the comment. We agree on the need to add the comparison with GNAN as it is also pointed out by other reviewers. Also, as highlighted by reviewer Vtmq, a comparison of the classification performances with the other SEGNNs was missing. We have also GNAN among them. We report the results in the table in the replies to reviewer Vtmq. However we want to highlight that the official code of GNAN does not allow for reproduction of the results provided in the paper. For this reason we provide  also the results from the GNAN paper to show that our model provides performances that are in line with other state of the art models.
>
> 3. **Missing citations.**   We will include proper support for the statement of line 118 by proposing to insert [1,15] as citations.
> 4. **Binarizing vectors holding a semantic meaning.** We understand the limitation in case of continuous input node features. However, the limitation is only present when input features do not have a semantic meaning. While in GloVe we agree that thresholding on input features is difficult to interpret, in chemistry this is not true: thresholding on the sum of atom radius in a certain neighborhood can actually be interpreted as whether the model is recognizing areas with either high or low "steric hindrance", a property that plays an important role in chemistry. The same applies for instance with the atom charge. However, we propose the following addition in the conclusions in line 366, to highlight it as a limitation: "... composed of multiple steps. Also, thresholding on input features might not be interpretable in cases where they are not attributed with specific semantic meanings. Future work could investigate case the integration with prototype-based or concept-based techniques  that could improve interpretability thanks to their case-based reasoning."
> 5. **Sigmoid Problems.** We realize our explanation may have been unclear. The issues related to the sigmoid activation arise in TELL due to its specific usage: the sigmoid is applied at multiple points, to binarize inputs, to constrain weights to be non-negative, and to produce the final output. These uses hinder gradient flow in multi-layer settings. In contrast, the sigmoid activation in a standard GIN, even when replaced with a Gumbel-sigmoid as in our pretraining phase, does not introduce the same optimization difficulties. Indeed, our black-box GIN model with sigmoid is stable and achieves performances aligned with prior works. Without pretraining, LogiX-GIN does not learn at all.
> 6. **Invalid pruning assumption.** When pruning, we do not look at layers as single entities. Layers are pruned one-by-one, in the sense that when pruning a layer, the parameters of the others are freezed. However, during pruning we keep cross-entropy to ensure that there is not degradation in the performances. In Table 2, indeed, we show that in some cases, performance even improve.
> 7. **Dataset diversity.** We perform evaluation on 10 datasets that are often used in prior work. Relating to the study of the network, we have already agreed that obtaining sematically valid rules on non-semantic continuous inputs is a limitation as would provide non-sematic interpretabtion. However, the model would still report rules that are perfectly matching its behavior, thus keeping its self-explainable nature.
> 8. **Spurious agreement between explanation and performance.**  We thank the reviewer for this comment. The reviewer's intuition would is correct. However alignment here is presented following Armgaan et al. [1] to ensure a fair comparison. In their work Armgaan et al. [1] consider alignment as the accuracy between the rules and the model predictions. They consider rules activations not using subgraph presence, but using activations of the symbolic regression over computation trees. With the same method we use rules activation on the logic extracted from the last TELL. As we understand that this is not properly explained we propose to improve its explanation and Figure 1 caption. Additional details are provided in replies to other questions of the reviewers and other reviewers.
> 9. **Too many rules to be considered interpretable.** We understand the concerns of the reviewer. Although there might be some problems regarding the semantic and subjective interpretability of the rules, in this work, we are more concerned on objective mathematical explainability given by the equivalence of the logic rules with LogiX-GIN's behavior.
> 10. **Instability.**  The high difference on seed variations highly depend on the pre-training strategy. Additionally, the number mentioned identify the amount of non-zero weights, and not the amount of rules. There could be weights that are not zero but still with low values that are not used in the rule.
> 11. **Ablations.** We did not include such ablations as they lead to inability of the model to learn. Regarding the removal of pre-training, the model does not learn at all. Regarding pruning layers individually, our strategy is justified by the fact that we do it from the last to the first. If we pruned the model as a whole we would actually end up in the  "invalid pruning assumption", as the removal of a weight at layer $L-1$ would critically impact layer $L$.
> 12. **Edge-features limitation.** We will include the impossibility of using edge features as a limitations.
> 13. **Figure 6 quality.** Although we cannot upload the figure for proof due to rebuttal regulations, we will modifiy the image according to the reviewer's comments.
> 14. **Minor corrections:**
>  - We thank the reviewer for the correction in line 140.
>  - We will increase the font of Figure 2. However, we cannot upload figures/links in the rebuttal.
>  - We thank the reviewer for the correction in Figure 10.
>  - The mentioned CPU only comes with 10 cores. However we will include the number of cores in section G.
>  - We agree with the reviewer and we will start the checklist on a new page.
> 15.  **Questions:**
>  -  **Non-negative constraint.** Following the results of Ragno et al. [19], we use a combination of the sigmoid and exponential functions: $W^+ = \exp{W_1} \odot \sigma{W_2}$ where $\odot$ is the Hadamard product. They find that this solution helps because the sigmoid acts as a switch for input selection while the exponential function scales the values to compensate the fact that the component of the sigmoid outputs only values between 0 and 1.
>  - **Input node features binarization.** Input node features are not binarized, we directly use LogiX-GIN, which using the Fourier threshold function learns intervals. In case of non-binary features, what happens is that the first layer of the network thresholds over the sum of the features in a certain neighbor.
>  - **Sigmoid binarization.** We use 0.5 as threshold. Also, as we replied in response to a comment of Reviewer LV2N, we failed to mention that we use a temperature scalar to enforce binary output values from the sigmoid and therefore proposed to update Eq. 7.
>
>  - **Pruning order.** Layers are pruned individually  and in reverse order because of the dependencies between the rules. Assuming layer $L$ uses some information from layer $L-1$ that are not necessary, we have to first prune them from $L$ and only then we can prune $L-1$, otherwise we will impact on the performances.
> - **Pretraining using GIN.** Only for pretraining, we use a GIN model that is trained with Gumbel-sigmoid activation function, thus having binary latent spaces.
> - **Alignment.** The reviewer stands corrected: the alignment it represents the accuracy between model's predictions and rules activations. We will specify this in Figure 1 caption.
> - **Other Layer-wise Rules and Negative Classes.** Unfortunately we cannot provide the figure in the rebuttal phase. We have only included the positive class of MUTAG for space issues, the image wouldn't enter in a paper space without affecting visibility and we thought about putting it in a notebook after publication. Despite this, to answer to your interest, in Ba2Motifs we have interestingly found that the models either had these two behaviors: (1) using rules to find nodes that belong to the loop and counting them (for the opposite class, it would count that it is under 5; (2) using rules to find nodes belonging to the BA distribution, then negating them to select the loops and finally count the nodes in the loop. This is the same in the case of BAMultiShapes, but it checks that the nodes belonging to the shapes are within well defined intervals to avoid returning 1 when all the three are present.
> - **Complexity and versatility of high level logic explanations.** By versatility we mean that it can work also when the model is bigger, whereas GraphTrail does not work in BAMultiShapes in our case. Also, 651 corresponds to non-zero weights, not to the rules. The amount of rules are instead shown in Figure 3.
> - **Figure 1 rules.** Figure 1 shows a different analysis and global rules on patterns, only using the last TELL layer. We acknowledge the limited clarity of the procedure used to extract the rules and for this reason we add an explantion of the procedure in the appendix. To give an intuition, we exploit the last TELL layer to extract rules that activate over aggregated outputs from the convolutional layers. For each rule we extract all the activating isomorphic subgraphs using Cordella et al. (2001).  Additionally, also in this case, it is important to remind that 829 and 651 are not the number of rules, but instead the number of non-zero weights.

---

> > ### Comment · Reviewer_Qw5B · 2025-08-06
> >
> > Thank you for engaging with the reviews. I have a few comments and questions before making my final remarks.
> >
> > **Weakness 2**: Thank you for conducting these additional experiments.
> >
> > **Weakness 8**: The response does not address the concern raised. The comparison with Armgaan et al.'s GraphTrail is misplaced. GraphTrail is as an explainer for black-box models, hence its fidelity (which the authors here refer to as alignment) is computed relative to that black-box model. In contrast, LogiX-GIN is as an inherently interpretable model. There is no separate black box; the rules are the model. In this case, alignment should refer to the match between the rule outputs and the ground truth labels, rather than with model predictions, since both the rule and the model are effectively the same entity.
> >
> > Regardless of this definitional nuance, Table 1 reports 100% accuracy for LogiX-GIN on BA-Multishapes, suggesting that it has perfectly captured the underlying logic. This raises a concern: the formula presented in Figure 1, as it stands, should not achieve perfect accuracy given the complexity of the task. This discrepancy casts doubt on the validity of other experimental results and warrants closer scrutiny.
> >
> > This point also connects to question 6 which asked:
> > > Is “alignment” the rule’s accuracy against the model’s predictions? And in LogiX-GIN, the model and rules are one and the same resulting in 100% alignment. Is my understanding correct?
> >
> > The authors responded:
> > > alignment represents the accuracy between model's predictions and rules activations.
> >
> > I'd like some clarification here. Since the rules and the model are the same in LogiX-GIN, what distinction is being drawn between them? Greater clarity on this point would help solidify the interpretation of alignment in this context.
> >
> > **Weakness 11**: The authors did not provide the ablation on pruning individual layers vs. pruning the model as a whole. If pruning is done at the model level, the loss function would jointly optimize all layers and should, in principle, preserve critical connections. I believe the requested ablation would have settled this. Also, please elaborate what the "invalid pruning assumption" is.
> >
> > **Final Explanations**:
> > In the response to Weakness 9 and the reply to reviewer LV2N (regarding the Global Explanation Procedure), it is mentioned that final explanations are obtained via isomorphism between subgraphs that activate certain weights. This introduces ambiguity. If the explanations are ultimately sets of isomorphic subgraphs, it is unclear how the logic represented by the model’s weights, one of the core contributions of the paper, is of use to the downstream user.
> >
> > What should an end-user do after training LogiX-GIN? Are the individual weights in the final layer meant to be interpreted directly? Should the user instead rely on the set of isomorphic subgraphs? Or a mix of the two, if so, how? Details on how an end-user is supposed to use LogiX-GIN would be quite helpful.
> >
> > **Question:** Is graded modal logic limited to the weights inside LogiX-GIN? Are the global explanations i.e., the logic learnt by LogiX-GIN, not output in that form? If yes, can you present those explanations for the datasets in the paper.

---

> > > ### Author Response · Authors · 2025-08-07
> > >
> > > We thank the reviewer for the time taken to analyze our replies and the feedback provided.
> > > - **Weakness 8 + Final Explanations**:
> > > We reply to both questions together as they are interconnected. The main reason we decided to compare with GraphTrail is that both methods yield explanations for GNN in the form of logic rules. However, our approach is different as the logic rules can be obtained for the whole set of layers. However, we still wanted to have a comparison, and, for this reason, we chose to focus on the last logic layer (the one after the readout) only for this experiment. Considering this last layer, we can extract rules on the activations coming from the readout. However, the rules that we obtain are still different from the ones of GraphTrail. While in GraphTrail the authors have rules on the presence of specific computation trees, our rules are activated when the output of the readout function falls within specific thresholds, therefore checking whether specific number of nodes have some features activated or not. In this scenario, we decided to offer a comparison following the same procedure of GraphTrail. What GraphTrail does is to check whether the activations of the logic rules on the computation trees reflect the output of the model. Following the same logic, we check whether the rules computed on the inputs of the last layer reflect the output of the model. In this case we need to use the model as reference, otherwise we wouldn't respect the procedure proposed in GraphTrail. What happens with BaMultiShapes is that the rules we get from our model perform exactly as the model, due to the fact that they count over node features and do not yield positive outcome if the three patterns are present. However, as the reviewer pointed out, the visualization for LogiX-GIN is a depiction of the graphs obtained using isomorphism to identify the cases where the rules are activated. Here, the reviewer is correct to say that this can introduce some approximation as different graphs could be isomorphic. However, it is important to mention that this problem is also present in GraphTrail, as multiple graphs could correspond to a computation tree. We think that the lack of clarity here is generated by the fact that the alignment is presented next to the figures (we call it alignment to highlight the distinction with the fidelity of node attributions). This brings to think that the alignment is measured with the presence of the highlighted subgraphs, while it is calculated on computation trees and activation of features, in GraphTrail and LogiX-GIN, respectively. Hoping that this is clearer, we can now discuss how the user should behave after training LogiX-GIN. The two types of explanations (global rules and layer-wise rules) offer two point of views on the behavior of the model. Layer-wise rules are the most complete explanation one could get as they return the perfect reasoning of the model. However, in highly complex tasks, it could be difficult to analyze the model if rules are numerous. For this reason, one could look at visual global rules, which offer a broader visualization that only focuses on the last layer.
> > > In truth, a third visualization could be obtained, using node attributions, that we present in the appendix for matters of space. To summarize, layer-wise rules show the full behavior of the model, global rules identify patterns and node-attributions point towards important nodes. With LogiX-GIN, all the three can be obtained. It is up to the end-user to choose the desired granularity.
> > >
> > > - **Weakness 11:** In the rebuttal we had limited space and we preferred to argument the reasons for pruning layer-by-layer. However, given the additional space of comments and the concern of the reviewer, we have now performed an ablation study on the BBBP and Ba2Motifs dataset. We report for each dataset, the percent decrease of non-zero weights (from non pruned to pruned) when using the full pruning and layer-by-layer pruning. In both cases, the layer-by-layer option better reduces the number of non-zero weights. Instead, jointly optimizing the weights leads to worse pruning due to the interdependencies of rules: if layer $\ell$ uses a feature of layer $\ell-1$, then it must be pruned first in the former and then in the latter. This is what we meant by "invalid pruning assumption", i.e, pruning weights that are needed in rules of subsequent layers. We actually used this term as it had already been used by the reviewer while referring to our pruning, to question whether it would take into account rule dependencies.
> > >
> > > | Dataset       | Layer by layer | Full |
> > > |-------------|-------|-------|
> > > | Ba2Motifs   | 83.13 | 78.84 |
> > > | BBBP        | 30.39 | 0.00  |

---

> > > > ### Author Response · Authors · 2025-08-07
> > > >
> > > > - **Question**
> > > > We are not sure whether we well understood the question, for this reason we will try to clarify some statements, hoping to address the concern. GML is the logic formalism that we use to express the logic rules for our model. This is due to the fact that we design our model to be directly convertible in GML formulas. Regarding "GML being limited to the weights", GML formulas are directly obtained from the weights. Finally, we already show the GML for a model on our datasets in Figure 2, where we report a visualization of the GML formulas on the MUTAG dataset that are directly obtained from the model weights. We hope this addresses the question of the reviewer. If not, we are open to better clarify the matter.

---

> > > > ### Comment · Reviewer_Qw5B · 2025-08-07
> > > >
> > > > Thank you for your response.
> > > >
> > > > > Following the same logic, we check whether the rules computed on the inputs of the last layer reflect the output of the model. In this case we need to use the model as reference, otherwise we wouldn't respect the procedure proposed in GraphTrail
> > > >
> > > > In GraphTrail, the blackbox is the "model" used as a reference. Who is that "model" here that the authors speak of?
> > > >
> > > > ---
> > > >
> > > > **Ablation:** Thank you for conducting the pruning ablation.
> > > >
> > > > ---
> > > >
> > > > > Layer-wise rules are the most complete explanation one could get as they return the perfect reasoning of the model. However, in highly complex tasks, it could be difficult to analyze the model if rules are numerous. For this reason, one could look at visual global rules, which offer a broader visualization that only focuses on the last layer.
> > > >
> > > > Thanks for the clarification. So the end-user has multiple options. I have another question. As indicated in lines 225-229 and later proven in appendix B, in theory the weights of the LogiX-GIN are convertible to graded modal logic (GML). Do I, as an end user, have access to those conversions, i.e., does LogiX-GIN's pipeline and codebase have that conversion baked into it where I just query and get the equivalent GML logic for any weight/layer I want? For example, the GML logic of the layers in figure 2. How does an end-user get those GML equivalents?
> > > >
> > > > ---
> > > >
> > > > > Question: Is graded modal logic limited to the weights inside LogiX-GIN? Are the global explanations i.e., the logic learnt by LogiX-GIN, not output in that form? If yes, can you present those explanations for the datasets in the paper.
> > > >
> > > > I'd like to connect this to the response from the authors to weakness 8:
> > > > > What happens with BaMultiShapes is that the rules we get from our model perform exactly as the model, due to the fact that they *count over node features and do not yield positive outcome if the three patterns are present.*
> > > >
> > > > This is what I mean by presenting logic learnt by LogiX-GIN in GML format. This should be BAMultishapes' global explanation, and other datasets should have similar ones rather than what is presented in the paper. LogiX-GIN should aim to provide this style of explanations.

---

> > > > > ### Author Response · Authors · 2025-08-07
> > > > >
> > > > > We thank the reviewer for the feedback. We really appreciate these questions driven by interest, regardless of the grade the reviewer would choose and the outcome of our submission.
> > > > >
> > > > > **In GraphTrail, the blackbox is the "model" used as a reference. Who is that "model" here that the authors speak of?**
> > > > > In our case the reference is LogiX-GIN itself, as the rules are explaining that model, while for GraphTrail we use the blackbox GIN model. We understand that the reviewer could question that the references are different. However, if we used the labels as reference, then the explanation evaluation would depend on the models' performance. In this case, instead, the alignment measures whether the explanations reflect the model they are derived from. We would also like to mention that having different reference models is a standard practice when evaluating self-explainable models (eg. as done in PiGNN and GNN).
> > > > >
> > > > > **Do I, as an end user, have access to those conversions, i.e., does LogiX-GIN's pipeline and codebase have that conversion baked into it where I just query and get the equivalent GML logic for any weight/layer I want? For example, the GML logic of the layers in figure 2. How does an end-user get those GML equivalents?**
> > > > > Yes, we derived those rules using a jupyter notebook, therefore the user would have access to it. The jupyter notebook is already available in supplementary but, in case of acceptance, we will provide tutorials in the github repository.
> > > > >
> > > > > **This is what I mean by presenting logic learnt by LogiX-GIN in GML format. This should be BAMultishapes' global explanation, and other datasets should have similar ones rather than what is presented in the paper. LogiX-GIN should aim to provide this style of explanations.**
> > > > > This type of explanation can be extracted by the model using the notebook we provide in the supplementary: the same plot of Figure 2 can be obtained for any dataset using the script. Indeed, the replies we gave regarding Ba2Motifs and BaMultiShapes are exactly based on the fact that we have analyzed those explanations, with the same notebook. Unfortunately the page limit and the “letter format” of the scientific paper is a limitation for showing such explanations that take much space: the nodes would have been too small and same goes for the text. For this reason we opted for MUTAG as we saw that the model was simply "shorter".

---

> ### Comment · Area_Chair_Rh2d · 2025-08-05
> **Ping**
>
> Dear Reviewer,
>
> The deadline for the author-reviewer discussion is approaching (Aug 8, 11.59pm AoE).
> Please read carefully the authors' rebuttal and engage in meaningful discussion.
>
> Thank you,
> Your AC

---

> ### Comment · Reviewer_Qw5B · 2025-08-08
> **Final remarks**
>
> Thank you for the clarifications. In light of my discussion with the authors, I am raising my score from **reject** to a **borderline accept**.
>
> I still find the novelty limited by NeurIPS standards. However, there is precedent for similar extensions. For instance, one of the works cited by the authors, LEN [4], has been employed similarly for GNN explanations in GLGExplainer (ICLR 2023).
>
> I found the experimental section in the paper unconvincing. However, the additional experiments, ablations, and clarifications provided in the rebuttal have positively swayed my assessment.
>
> That said, I still find the alignment experiment quite odd and misleading, and I don’t think it adds value to the paper.
>
> I appreciate the authors’ acknowledgment of certain limitations, which helps clarify the scope of the work.
>
> Above all, the paper most certainly needs a detailed methodology subsection which demonstrates extraction of explanations in the form of graded modal logic, as that is THE selling point of the paper. While the authors have indicated that this is implemented in code, it is essential that it be presented explicitly in the paper.
>
> In its current form, which parts of the pipeline are intended for interpretability and how remains a bit ambiguous to me. There are multiple potential candidates and their interplay: weights, GML equivalents, activations, and isomorphic subgraphs. The paper would benefit from describing the intended use of each these aspects of the model’s output from an end-user's perspective. The authors have partially addressed this in the rebuttal, and incorporating that clarification into the paper would aid future readers.
>
> Finally, I thank the authors for the constructive and collegial exchange.

---

### Official Review · Reviewer_Vtmq · 2025-06-23

**Clarity:** 1
**Significance:** 2
**Originality:** 3
**Rating:** 2
**Confidence:** 4

**Summary:**

This paper combines logical explanation methods with self-explaining models by proposing a novel self-explaining graph neural network, **LogiX-GIN**. The model aims to automatically generate interpretable rules that adhere to *Graded Modal Logic (GML)*, while maintaining predictive performance and offering faithful, transparent explanations for its decisions.

**Questions:**

1. Consider including a diagram of the LogiX-GIN pipeline to illustrate data flow, binarization, distillation, and rule generation.
2. Please reorganize the notation system and add a comprehensive notation table.
3. Provide concrete examples of GML formulas to improve accessibility.
4. The logic extraction seems to focus on input feature–output feature relationships. Yet, the experiment on “faithful global logic rules” presents node-level rules. How is this abstraction from feature-level to node-level achieved?

**Ethical Concerns:**

["NO or VERY MINOR ethics concerns only"]

**Limitations:**

Yes

**Quality:**

1

**Strengths And Weaknesses:**

## Strengths

* While most prior work on logical explanation methods has focused on post-hoc techniques, this paper explores their integration into self-explaining models.
* The proposed LogiX-GIN model provides interpretable explanations across multiple datasets without sacrificing predictive accuracy.

## Weaknesses

1. **Unsupported Claims**:
   Several claims lack proper justification or references.

   * Lines 117–119 state that LENs and instance-level post-hoc methods are not faithful, but no evidence or citations are provided.
   * Lines 300–302 assert that LogiX-GIN highlights aromatic carbon structures as mutagenic, in contrast to GraphTrail’s NO₂ detection. However, the mutagenicity of simple aromatic rings is not well-established without specific functional groups like NO₂ or NH₂. These claims require citations or qualification.

2. **Inconsistent and Unclear Notation**:
   The paper lacks a notation table, and many symbols are ambiguous.

   * In Line 159, the equation \$y = \sigma(XW^+ + b)\$ uses \$X\$ and \$y\$ as vectors but with inconsistent capitalization. Since vectors are typically column vectors, the multiplication \$XW^+\$ is dimensionally incorrect—it should be \$y = \sigma(W^+x + b)\$ if \$x\$ is a column vector.
   * Similar notation issues occur in Equation 7 and Line 170, where \$\tilde{W} \in \mathbb{R}^I\$ should be lowercase to avoid confusion with matrices.
   * The semantics of \$E(x, y)\$ and the \$\models\$ symbol are not defined.

3. **Logical and Mathematical Errors**:

   * The statement \$y^{\text{bin}}\[j] = 1 \Leftrightarrow \sum\_{i \in S} W^+\[i, j] > -b\[j]\$ omits the dependence on input \$x\$, contradicting the earlier expression \$y = \sigma(XW^+ + b)\$.
   * Line 169’s \$\tilde{x} = \sigma(X \odot \exp(\tilde{W}^+ \tilde{b}))\$ does not logically lead to the binarization formula in Line 171, and \$\tilde{W}^+\$ is undefined.
   * Lines 209–210 incorrectly state that the aggregated nodes \$N(v) \cup {v}\$ correspond to \$\epsilon=1\$ in Equation 2, when they actually match \$\epsilon=0\$.

4. **Unclear Formalism**:

   * Lines 183–185 introduce GML formulas but use recursive expressions like \$\psi(x), \psi\_1(x), \psi\_2(x), \theta(y)\$ without explaining what constitutes a GML formula.
   * Similarly, Lines 186–192 use the \$\models\$ symbol in both definitions and explanations without clarification.
   * The training procedure in Section 4 is under-specified, including a lack of clarity on the loss function and the distillation mechanism.

5. **Limited Experimental Validation**:

   * The experimental comparison only includes standard GIN and LogiX-GIN. Other self-explaining GNNs—such as VGIB, KerGNN, and Pi-GNN, all mentioned in the paper—should also be included for a more comprehensive evaluation.

---

> ### Author Rebuttal · Authors · 2025-07-27
>
> We thank the reviewer for the time and the thorough evaluation. We will address the concerns hoping to clarify and further improve the quality of our method
> 1. **Unsupported Claims:**
>     - **GLGExplainer faithfulness.** The unfaithfulness of GLGExplainer (which demands for a post-hoc explanation to be run first and on LENs) was demonstrated by Armgaan et al. [1] while the unfaithfulness of LENs was demonstrated by Ragno et al. [19]. We acknowledge the lack of citations pointing to evidence and we propose to put [1,15] after the assertion in line 119.
>     - **Aromatic carbon structures as mutagenic.** We apologize for the lacking citation. Aromatic structures are well known to be involved with mutagenicity from Debnath et al. [8] and was also reported in Armgaan et al. [1]. To fix the lack of citations we therefore propose to put [1,8] after the assertion in line 302.
> 2. **Inconsistent and Unclear Notation:**
>     - We propose to the modify the defintion of $X$ in line 159 as follows "$X\in\mathbb{R}^{1\times I}$ to remove the doubt of whether the input is a row or a column vector. We indeed referred to it as row vector, given that generally in ML and DL, they are row vectors.
>     - We propose to correct the definition of $\tilde{W} \in \mathbb{R}^{1\times I}$ in Equation 7 and 170, following the same idea.
>     - We propose to insert the notation for $E(x,y)$ by modifying the sentence in line 178 with "... a symbol $E(x,y)$ for a binary relation between representing adjacency between nodes $x$ and $y$ of an undirected graph..." and the definition of the symbol $\models$ with with the edit in lines 186:
>
> "Given a graph $G$ and a node $v$, the relation $G, v \models \varphi(x)$ that indicates that $v \in G$ satisfies a generic GML formula $\varphi(x)$ is defined as follows:
> - $G, v \models P(x)$:
> node $v$ satisfies the atomic predicate $P$ (i.e., $v$ belongs to the unary predicate $P$),
> - $G, v \models \neg \psi(x)$:
> node $v$ does not satisfy the formula $\psi(x)$ (negation),
> - $G, v \models \psi_1(x) \land \psi_2(x)$:
> node $v$ satisfies both subformulas $\psi_1(x)$ and $\psi_2(x)$ (logical conjunction),
> - $G, v \models \psi_1(x) \lor \psi_2(x)$:
> node $v$ satisfies at least one of the subformulas $\psi_1(x)$ or $\psi_2(x)$ (logical disjunction),
> - $G, v \models \exists^{\geq N} y\, (E(x, y) \land \theta(y))$:
> node $v$ has at least $N$ neighbors $u$ such that $(v, u) \in E$ and each $u$ satisfies $\theta(y)$ (quantified neighborhood condition)."
>
> 3. **Logical and Mathematical Errors:**
>     - The statement $y^{\text{bin}}[j] = 1 \Leftrightarrow \sum_{i \in S} W^+[i, j] > -b[j]$ is a conclusion that was already mathematically demonstrated by Ragno et al. [19] (TELL) and does not constitute our contribution. Indeed we report it in the background section of our paper, being a contribution of a previous work. Nonetheless, Ragno et al. [19] show that the equation is true thanks to the fact that inputs are binary and weights are non-negative, giving the possibility of obtaining rules without depending on the input $x$.
>     - The expression in 169 is reported with a typo (indeed in the question, the reviewer has tried to fix the typo by adding a final parenthesis). The correct form is $\tilde{x} = \sigma(X \odot \exp(\tilde{W})+ \tilde{b})$. Also this is a result of Ragno et al. [19] and is presented in the background as it does not constitute our contribution.
>     - We thank the reviewer for the correction about $\epsilon$. The reviewer is correct and our work corresponds to $\epsilon$ = 0, we will correct it.
>
> 4. **Unclear Formalism:**
>     - **Inductive definition of GML.** We apologize if the presentation was not immediately clear. The use of recursive expressions reflects the inductive nature of the definition of GML formulas. This recursive style is standard in formal logic, where formulas can be nested and composed using logical operators. Each formula on the right-hand side of the grammar rule is itself a GML formula, defined by the same inductive process.
>     - **Undefined symbol.** Regarding the use of the satisfaction symbol $\models$, this follows standard logical notation: $a \models b$ means that the structure or element $a$ satisfies the formula or condition $b$. Nonetheless, to improve clarity for readers who may be less familiar with this convention, we propose the edit in lines 186 as already specified.
>
> 5. **Limited Experimental Validation:** We agree with the reviewer with the necessity to include the comparison in classification performances with other SE-GNNs. We didn't include them in the paper as we were more concerned about the loss of performance of LogiX-GIN w.r.t GIN. However, we will add in the appendix the following comparison table where we also include GNAN, as asked by another reviewer. However we want to highlight that the official code of GNAN does not allow for reproduction of the results provided in the paper (as also pointed out in an issue in their public github repository). For this reason we provide here also the results from the paper to show that our model provides performances that are in line with other state of the art models, with the additional capacity of providing full logic rules for its behavior.
>
> | Model     | Ba2Motifs          | BaMultiShapes        | MUTAG               | Mutagenicity        | NCI1                | PROTEINS            | BBBP                |
> |-----------|--------------------|----------------------|---------------------|---------------------|---------------------|---------------------|---------------------|
> | LogiX-GIN |  100.00 ± 0.00     |  100.00 ± 0.00       |   82.63 ± 5.58      |  79.31 ± 1.25       |  78.93 ± 1.51       |   72.05 ± 5.77      |  85.90 ± 0.99       |
> | PiGNN     |  99.89 ± 0.31      |  85.40 ± 5.08        |   82.51 ± 10.48     |  82.39 ± 1.68       |  78.54 ± 2.74       |   70.00 ± 2.44      |  83.54 ± 0.37       |
> | GIB       | 100.00 ± 0.00      |  97.60 ± 2.06        |   90.53 ± 6.14      |  80.14 ± 0.98       |  78.15 ± 1.32       |   68.47 ± 5.00      |  84.78 ± 2.57       |
> | KerGNN    | 98.80 ± 0.75       |  83.20 ± 1.94        |   82.11 ± 9.18      |  73.32 ± 2.93       |  69.00 ± 1.38       |   72.97 ± 4.87      |  84.12 ± 2.09       |
> | GNAN      | 49.10 ± 0.66       |  49.10 ± 0.66        |  55.79 ± 16.43      |  56.05 ± 1.49       |  50.80 ± 1.15    |   57.67 ± 2.62      |  41.22 ± 26.05       |
> |GNAN (From original paper) |       -          |         -            |        -            |  72.20 ± 1.00       |  76.90 ± 1.2        |   73.20 ± 3.10      |        -            |
>
> 6. **Questions**
>     - **Diagram of the LogiX-GIN pipeline to illustrate data flow, binarization, distillation, and rule generation.** Unfortunately we cannot include any image in the rebuttal, however we have created a diagram that we put in the appendix and in the final repository.
>     - **Please reorganize the notation system and add a comprehensive notation table.** We thank the reviewer for the suggestion. Space constraints in the main text do not allow us for a table, however, we will include such Table in the appendix.
>     - **Provide concrete examples of GML formulas to improve accessibility.** We have improved our presentation of the GML as already replied. However, this is an interesting idea, we will use the appendix to provide such examples together with the notation table.
>     - **How is this abstraction from feature-level to node-level achieved?** We have realized, also thanks to other reviewers, that the global patterns extraction was not well detailed. We have therefore decided to add an appendix section for that. However, since we do not have space here to present it, we will give you an intuition. The visual rules are extracted from the last tell layers, which uses activations of the convolutional layers to predict the output class. Since this last layer is a TELL, we use TELL extraction procedure for rules. Successively, for each rule, we extract from the training set all the isomorphic subgraphs (using using Cordella, L. et al. (2001)) that activate for such rules.

---

> ### Comment · Area_Chair_Rh2d · 2025-08-05
> **Ping**
>
> Dear Reviewer,
>
> The deadline for the author-reviewer discussion is approaching (Aug 8, 11.59pm AoE).
> Please read carefully the authors' rebuttal and engage in meaningful discussion.
>
> Thank you,
> Your AC

---

### Official Review · Reviewer_LV2N · 2025-07-02

**Clarity:** 1
**Significance:** 3
**Originality:** 3
**Rating:** 4
**Confidence:** 4

**Summary:**

The paper tackles the challenge of providing explanations for GNNs by proposing a new ante-hoc method called LogicX-GIN. The proposed model incorporates some constraints in the way information is processed by the architecture, making the model amenable to rule extraction. Claims are tested on several datasets, showing that the model retains the predictive power of black-box GNNs while allowing for extracting graded model logic formulas.

**Questions:**

- **Q1**: What is the role of the clamp function in Equation 5? I was expecting $\tilde{\alpha}$ scores to be binary in $\{0,1\}$, but it seems that $\tilde{\alpha}$ can take other values, like $0.1$. Why is it done in this way?

- **Q2**: Are $\tilde{\alpha}$ vectors or numbers? $\alpha$ is defined in line 209 as the sum of embedding vectors, but in Eq. 6, you write $\tilde{\alpha}=1$, which indicates they are simple scalars. Maybe you mean $\tilde{\alpha}[j]=1$, where $j$ is a single hidden dimension?

- **Q3**: Why is the additional Hoyer loss applied as a post-training step, rather than being added as an additional loss during training?

- **Q4**: Can you please elaborate on lines 336-337? It is not clear what the underlying message the authors are trying to convey.

- **Q5**: How does the model cope with graphs where node features are complex embeddings, like textual embeddings?

**Ethical Concerns:**

["NO or VERY MINOR ethics concerns only"]

**Final Justification:**

The authors did make a substantial attempt at addressing my concerns, which made me increase my score to weak accept.

Below is a justification of why I'm not in support of full acceptance.

Despite the clarity of the proposed contribution being increased after the authors' clarifications, some major concerns still apply. As also observed by other reviewers, the paper struggles in providing unambigous evidence that the final formulas are indeed more interpretable (see Qw5B), and the fact that the proposed approach requires both a pre-training (otherwise the model does not train properly) and a post-training (otherwise explanations are overly complex and less interpretable) makes the overall model unstable and arguably not easy for usage in new downstream tasks.

**Limitations:**

yes

**Quality:**

2

**Strengths And Weaknesses:**

# Strengths

- Investigating novel paradigms for self-explainable GNNs is a very interesting direction, and adapting logic explanations to the ante-hoc setting is rarely studied

- The paper makes a connection between the logical expressivity of GNNs and their relative logical explanations, which is a novel contribution

- Albeit the description of the proposed contribution is unclear in some parts (see below), the architecture remains relatively simple and does not require articulated components or training losses, which is a pro to the proposed method


# Weaknesses

- The model is grounded on the TELL architecture, but this is only briefly explained in the Background. The self-containment of the paper would benefit from providing more context and background about TELL. For example, it is not clear how the thresholding of input features works, requiring the reader to read the original paper to fully understand it.

- The architecture is presented as a sort of extension of the GIN architecture, but in practice, the aggregation function in Equation 4 shares only the sum over neighbors with the original GIN, making the connection not particularly strong. For example, GIN aggregates the self-loop by multiplying the $\epsilon$ parameter, which is neglected in LogicX-GIN by setting $\epsilon=1$. I believe the architecture can be presented in more general terms without grounding on GIN, but by referring to any GNN architecture with sum aggregation.

- The proposed solution, in particular Equations 5 and 6, is introduced abruptly. The clarity of the proposed solution would greatly benefit from more intuition and explanation regarding it. I also checked Appendix A, but it is not enough to have a feeling of the functioning of that operation. I think that providing some toy analytical examples would be useful to understand how this counting-based step function works.

- There is a mismatch between practice -- Eq. 5 in particular -- and the theory in Proposition 1. The proof is based on the fact that summing over $h_u^k$ yields, for each hidden dimension $j$, the count of neighbors where $h_u^k[j]=1$. Note that this is true as long as each $h_u^k[j]$ is binary (taking values either 1 or 0). However, according to Eq. 7, each $h_u^k[j]$ is not necessarily binary, as the sigmoid function can yield any value in between 0 and 1.

- There is a conceptual contradiction regarding the pretraining stage. The authors claim that the sigmoid function of LogicX-GIN can create issues with model training, and therefore propose to train a standard GIN model to "initialize" model weights. Nonetheless, this standard GIN is still implemented with a sigmoid activation function, which supposedly inherits the same problem of LogicX-GIN (since they both use sigmoid activations). Even more, the standard GNN employs a Gumbel-sigmoid activation, which arguably makes the training even more unstable. Then, is it really the sigmoid activation function the problem when training LogicX-GIN from scratch?

- The paper lacks the fundamental part of explaining how explanations are extracted from LogicX-GIN. The authors broadly refer to *the activations of the literals after the readout function can be interpreted as indicative patterns whose presence directly affects the final prediction* in line 292. Nonetheless, this is not trivial, and being part of the key contribution of the paper, it is definitely not explained well enough.



# Minors

- In line 1: Graphs do not necessarily *require* GNNs. There exist many alternatives to GNNs, like Graph Transformers and kernel-based approaches. Please update this line.

- Given that the paper proposes to move beyond the usual paradigm of topological explanations, it might be worth including SE-GNNs that are not intrinsically topological, like [1,2,3]. This would make the related work section more complete.

- Line 119: Please add some references to support the claim that post-hoc methods are not properly faithful.

- Line 148: The statement: *The sum aggregation can distinguish between different neighborhood structures without ambiguity* is not correct. In fact, albeit expressive, GIN still has not reached full discrimination power as it is still upper bounded by 1-WL (see GIN's paper).

- Line 157: *TELL is designed to be interpretable in terms of logic* sounds a bit vague. Please reformulate this statement to make it more precise.

- Line 157: What do you mean by *feed-forward operation*? Is this a linear layer?

- Equation 3: What is the difference between any $\psi_i(x)$ and $\theta(y)$? If they are simply different GML formulas, I would rather make the notation uniform and replace $\theta$ with $\psi_3$.

- It is not clear whether the statement *which is necessary to represent counting operations in GML* in line 210 refers to the sum aggregation or to the fact that $\epsilon=1$.

- The caption of Figure 1 is not self-contained. Details should be added, like what is *alignment* (a proper formal definition is lacking even in Sec. 5), and whether those explanations are global or local.



[1] Graphchef: Learning the recipe of your dataset. IMLH 2023


[2] The intelligible and effective graph neural additive network. NeurIPS 2024


[3] Beyond Topological Self-Explainable GNNs: A Formal Explainability Perspective. ICML 2025

---

> ### Author Rebuttal · Authors · 2025-07-27
>
> We thank the reviewer for the thorough work. We in the following we address the raised concerns raised by the reviewer, hoping to improve their concerns regarding quality and clarity.
>
> 1. **TELL presentation and thresholding function.**
>     Due to space limits, we opted for a concise presentation of background work like TELL. However, we acknowledge the need for clarity, especially for the thresholding. Thanks to a comment from another reviewer, we found a typo in the thresholding eq. that impacted readability. We have corrected it in 169 to match the correct formulation, which the same from Ragno et al. [19]:
> $$
> \tilde{x} = \sigma(X \odot \exp(\tilde{W}) + \tilde{b}).
> $$
>     That said, we use a different thresholding to operate the counting semantics of GML. This is described in Eq. 5 using a novel learnable Fourier step function. To improve clarity in the revised paper, we rewrite lines 211–214 as follows: “To apply logic-based rules, we binarize them using a novel thresholding mechanism. Specifically, we need a function capable to activate if its input falls within certain intervals. For this reason, we design a parametric Fourier step function, which flexibly and differentiably activates (outputs one) when counts fall within learnable value ranges.”
> 2. **GIN Extension.**
>     We acknowledge that we use a "restricted" form of GIN. However, referring to our approach as a GNN architecture with sum aggregation would be too broad. We believe the link to GIN to be meaningful, especially since in Xu et al. [26], the authors show that  $\epsilon = 0$ can in fact improve the performances. As pointed out by another reviewer, there was a typo: we actually use $\epsilon = 0$. This is also consistent with the default configuration in PyG, with $\epsilon=0$ and not trainable.
> 3. **Sigmoid binary values.**
>     The observation is accurate: Eq. 7, as currently written, applies a standard sigmoid function, which yields continuous outputs in  $[0,1]$, and this would invalidate the counting assumption used in Prop. 1 if left unaddressed. In practice, however, we overcome this issue by using a temperature term: we divide the input of the sigmoid by a small $\tau$ (set to $\tau = 10^{-4}$) to have outputs very close to 0 or 1. This was not explicitly mentioned in the paper. We have now corrected Eq. 7 to make this explicit:
>     $$h^{(k)}_v = \lambda^{(k)}\left( \tilde{a}^{(k)}_v \right) = \sigma\left( \frac{ \tilde{a}^{(k)}_v W^{+(k)} + b^{(k)} }{\tau} \right), \quad \text{with } \tau = 10^{-4}    $$
> 4. **Problems of the sigmoid.** We realize our explanation may have been unclear. The issues related to the sigmoid activation arise in TELL due to its specific usage: the sigmoid is applied at multiple points, to binarize inputs, to constrain weights to be non-negative, and to produce the final output. These uses hinder gradient flow in multi-layer settings. In contrast, the sigmoid activation in a standard GIN, even when replaced with a Gumbel-sigmoid as in our pretraining phase, does not introduce the same optimization difficulties. Indeed, our black-box GIN model with sigmoid is stable and achieves performances aligned with prior works. Without pretraining, LogiX-GIN does not learn at all.
> 5. **Global Explanation Procedure.**
>    We understand that the procedure to obtain the patterns in Figure 1 was not properly explained, we then propose to add a new Appendix section. In line 293, we insert the following sentence: "Further details on the procedure to extract global rules are available in Appendix G". Additionally, we will introduce the following explanation in Appendix G. For spaces constraints in the rebuttal we cannot include the full section. However, to give an idea, we use the TELL rule extraction system on the last layer to find the nodes that activate logic rules. Then, we extract isomorphic subgraphs using Cordella, L. et al. (2001) to determine visual rules.
> 6. **Minors:**
>     - **Graphs do not need GNNs.** We agree with the reviewer and we will change to "Graphs are complex, non-Euclidean structures that require specialized models, such as Graph Neural Networks, Graph Transformers, or kernel-based approaches, to effectively capture their relational patterns."
>     - **Non-topological SEGNNs are not cited.** We propose the following two edits: (1) Lines 35-37 "However, existing SE-GNNs present some limitations: topology-based approaches [14, 5, 18, 32, 17] identify the most relevant subgraphs while providing little insight into the logical rules underlying how the model represents graphs and makes its decisions [15, a]; symbolic approaches propose distillation of the models into simpler or more interpretable architectures such as decision trees; and other approaches restrict the focus on node features, without using information about substructures [b,c]"; (2) In line 122-125, "Taking a different direction, Pluska et al. [15] and Muller et al. [a] propose distilling symbolic models from a GNN using decision trees. Despite this difference, their methods are either post-hoc, as the rules are extracted from graph embeddings, or the final models is are longer a GNN and cannot be reused in other neural network settings, thus losing the advantages of neural representations";
>     [a] Graphchef: Learning the recipe of your dataset. IMLH 2023
>     [b] The intelligible and effective graph neural additive network. NeurIPS 2024
>     [c] Beyond Topological Self-Explainable GNNs: A Formal Explainability Perspective. ICML 2025
>     - **Missing Citations.** We acknowledge the lack of citations pointing to evidence and we propose to put [1,15] after the assertion in line 119.
>     - **WL test.** We propose the following edits on lines 147-151: "The sum aggregation can distinguish between different neighborhood structures and the MLP update function provides the necessary expressive capacity to obtain meaningful representations, enabling GIN to approximate the 1-WL (Weisfeiler-Lehman) graph isomorphism test, which is a strong baseline for distinguishing non-isomorphic graphs."
>     - **TELL is designed to be interpretable in terms of logic.** We understand that the sentence might not be clear, for line 115, we propose the following edit: "TELL is designed to be interpretable through a direct conversion into first-order logic formulas."
>     - **Feedforward operation.** The reviewer is indeed right, we can say it is a linear layer with a non linear activation function. Thanks for pointing it out. Indeed converting it to "a standard linear layer" is more appropriate.
>     - **GML formula.** We opted for $\theta(y)$ as it is applied to neighboring nodes. However the reviewer stands corrected, changing it to $\psi_3(y)$ does not impact the correctness of the definition. We will follow the suggestion of the reviewer and change it to avoid reading ambiguities.
>     - **Sentence unclarity.** The outlined sentence is referred to the aggregation scheme. However, to increase readability we propose the following change in line 209: "This aggregation is equivalent to the one of GIN when setting..."
>     - **Figure 1 Caption.** We agree with the reviewer that Figure 1 deserves a more detailed caption. For this reason, we propose the following new caption:
>
>    "Visual comparison between rules extracted with GraphTrail and LogiX-GIN. For LogiX-GIN, the global rules on subgraphs are obtained from the last classification layer following the same procedure of Ragno et al. [19]. Alignment is measured as the accuracy between the predictions of the model and the logic activations of the rules following  Armgaan et al. [1]. Additional details about the explanations extraction are provided in Appendix G."
>
>
> 7. **Questions:**
>     1. **Role of the clamp.** As visible in Figure 4, the fourier step function, when applying low $\tau$ values, might have oscillating values around one (very close to 1). While we can afford having values that are slightly lower than 1, in case they are slightly bigger than one we would get out of the [0,1] interval. However this does not impact the interpretability as we use a very high $\tau$ values.
>     2. **$\tilde{\alpha}$ are vectors?** They are actually vectors. Indeed, as suggested by the reviewer, we do mean $\tilde{\alpha}[j]=1$, thanks for the correction. We will edit the sentence as suggested.
>     3. **Hoyer loss only in pruning?** During training we follow a classical approach using L2 regularization to ensure smooth learning. During pruning, instead, the Hoyer Loss applies a strong regularization that we only want to apply once the model is trained, in order to avoid training instabilities.
>     4. **XAI Evaluation** Given the counting operation, GNNs might learn that a certain class is determined if there exist at least n nodes that have a certain embedding activated, as in the case of Figure 2. Therefore, if we use soft input attributions and we remove nodes one by one, as long as we stay within the range, the prediction does not change. This is actually important to keep in mind because most of the evaluation mechanisms for post-hoc approaches remove the most important nodes up to certain sparsity levels and evaluate prediction changes.
>     5. **Complex embeddings, like textual embeddings?** Following the suggestion of another reviewer we have decided to mark this as a limitation as it would be impossible to interpret the logic rules on "non-interpretable embeddings". The model would be still convertible into rules, however we the input literals wouldn't be interpretable. We leave to future work the exploration of other techniques to augment such interpretability through concepts or prototypes.

---

> > ### Comment · Reviewer_LV2N · 2025-08-04
> >
> > Thank you for your clarifications. Overall, every point I raised was addressed.
> >
> > I'll increase my score from 3 to 4, weak acceptance. Below is a more detailed justification of why I'm not in support of full acceptance.
> >
> > Despite the clarity of the proposed contribution being increased after the authors' clarifications, some major concerns still apply. As also observed by other reviewers, the paper struggles in providing unambigous evidence that the final formulas are indeed more interpretable (see Qw5B), and the fact that the proposed approach requires both a pre-training (otherwise the model does not train properly) and a post-training (otherwise explanations are overly complex and less interpretable) makes the overall model unstable and arguably not easy for usage in new downstream tasks.

---

> > > ### Author Response · Authors · 2025-08-05
> > >
> > > We thank the reviewer for their response. We acknowledge the necessity of both pre-training and post-training; however, these stages do not interfere with the potential application of the model in downstream tasks. The final model remains a GNN composed entirely of fully differentiable layers. Pre-training serves as a way to "pre-initialize" the model and can be seen as analogous to fine-tuning from a related model. Post-training, on the other hand, functions as a form of strong regularization.
> > >
> > > Regarding the comment on the formulas being "more interpretable," we are unsure of the intended point of comparison. The size of the learned rules is comparable to those produced by post-hoc methods (e.g., GraphChef), with the important distinction that our rules come with a mathematical guarantee of exact alignment with the model, rather than being approximations of it.

---

### Official Review · Reviewer_Dowp · 2025-07-02

**Clarity:** 3
**Significance:** 2
**Originality:** 2
**Rating:** 5
**Confidence:** 4

**Summary:**

The authors introduce an inherently explainable GNN method. They use transparent explainable logic layer and graded modal logic to make an interpretable version of a GIN layer. This interpretable model is shown to have similar performance to the black-box GIN while extracting sensible logic rules for real-world datasets commonly used to evaluate explainable GNNs.

**Questions:**

In Node Attribution section, you state that attributions of LogiX-GIN are taken from the rules activated in the last layer. Is it instead possible to provide an end-to-end node attribution in your method, taking all layers into account? Something akin to running integrated gradients on the method. While I appreciate the value of having inspectable model rules, in practice, having good node attribution can be very useful for the end-user when inspecting a given graph.

Maybe to get this 'full' explanation one could also just expand the logic rules of all the layers into one? Or does that in practice become unreasonably large and is unsimplyfiable?

One missing reference is GraphChef: https://openreview.net/pdf?id=IjMUGuUmBI which in some ways is quite similar: inherently explainable architecture based on GIN, with a transfer learning step from a black-box GIN, but uses decision trees instead of logic rules (thus not differentiable).

**Ethical Concerns:**

["NO or VERY MINOR ethics concerns only"]

**Final Justification:**

After rebuttal and reading other reviews I still like the paper and recommend acceptance.

**Limitations:**

Yes

**Quality:**

3

**Strengths And Weaknesses:**

I think work on self-explainable GNNs is super important, as it greatly increases the trustworthiness of the methods (which post-hoc analysis can't necessarily provide). The proposed method in my opinion is a valuable contribution to this sub-field. As far as I know this is first method providing a self-explainable GNN of this type that uses logic rules. The fact that the final self-explainable method uses TELL layers is quite handy, as the model still is GPU accelerated and in principle differentiable (even though the gradients might be bad due to sigmoids). Which contrasts with some other self-explainable GNNs that exist.
The writing and presentation are clear. The benchmarks used are quite standard and the results look good.

The authors themselves acknowledge the two main weaknesses that I also see: 1) needing to start with a black-box GIN, which I think is a very minor issue and 2) that to extract explanations they rely on logic rules in the last layer. Here I think the method would be noticeably improved if authors provided a way to easily extract the full explanation that takes into account all layers. Of course this information is still in the model, so persistent user could pull it out manually, but it would be quite cumbersome.

---

> ### Author Rebuttal · Authors · 2025-07-27
>
> We thank the reviewer for the comments and suggestions. Regarding the second weakness, only the "global" explanations of Figure 1 are w.r.t. the last layer. Instead, the innovation of our proposal is exactly that, with LogiX-GIN, we can extract full rules for all the layers, as shown in Figure 2.
>
> For the first question, although the node attributions use the last layer, they are still generated using the activations that come from the first layers.
>
> For the second question, it is in theory possible. However, we believe that merging the explanations would make it even less interpretable, given that other reviewers have already raised concerns about the interpretability of logic and the amount of rules.
>
> Regarding question 3, we have included GraphChef as suggested by the reviewer and also other reviewers.

---

### Author Response · Authors · 2025-08-04

Dear reviewers, we have not received any feedback regarding our replies to your constructive comments. We believe we have done our best to clarify the definitions and our procedure, and, more importantly, we have performed the requested experiments, which confirmed the quality of our approach. With this message, we do not intend to be impolite, but rather to ask for your feedback in case you believe the work is still not valuable enough for the conference, or to kindly reconsider your grade.

---

### Note · Authors · 2025-08-12

We thank the Reviewers and ACs for their time, effort, and constructive feedback.
Our work introduces a novel GNN that can be directly transformed into logic rules, uniquely combining full differentiability with faithful, layer-wise explanations, without relying on surrogate models such as decision trees or other post-hoc methods.

The reviews highlighted missing citations, evaluations, and clarifications. We addressed these points by incorporating the requested references, performing extra experiments (which further support our approach’s quality), and revising the manuscript for clarity.
We are pleased that 2 reviewers engaged in the discussion and updated their scores upward. In these remarks, we address the remaining points and invite the reviewers who did not participate to reconsider our responses.

**Reviewer Dowp** initially recommended acceptance and raised questions on the explanation extraction process that we believe to have fully addressed.

**Reviewer LV2N** confirmed that all raised points were addressed and moved to bordeline accept but noted two residual concerns:
1. *Lack of evidence that our formulas are more interpretable.* As the reviewer did not provide a reference to assess "more interpretability", we emphasize that our model uniquely produces logic formulas mathematically proved to match each model layer, with no approximation.
2. *Pre-training and post-training limit downstream applicability.* In response to this, we reaffirm that LogiX-GIN is fully differentiable, and these steps do not impede use in downstream tasks.

**Reviewer Qw5B** participated in a constructive exchange, also moving to bordeline accept, and suggested integrating our clarifications into the paper. The reviewer also noted a concern regarding the novelty of our work, pointing towards similar approaches (LEN) applied to GNNs (GLGExplainer). We argue that a GNN layer that can be directly converted into logic rules, without any form of approximation or dependence on post-hoc methods, is entirely novel. LENs require post-hoc rule extraction and have only been used post-hoc for GNN explanations. In GLGExplainer, PGExplainer is first applied to obtain node attributions, after which LENs generate rules.

**Reviewer Vtmq** did not participate in the discussion. While we note this without criticism, their concerns overlap with those of LV2N and Qw5B, both of whom confirmed that they were addressed. We respectfully ask Reviewer Vtmq and the ACs to consider this.

---

### Decision · Program_Chairs · 2025-09-17

**Decision:**

Accept (poster)

**Comment:**

Most reviewers appreciated the aims and technical contributions of the paper. The author-reviewer discussion was mostly productive, and the authors managed to clarify many of the more serious issues raised by most the reviewers. Residual concerns about incrementality, interpretability and usability are comparatively modest. This makes me lean towards acceptance. (One reviewer was very critical of the work and raised sensible points, but was completely unresponsive to the author's on-point rebuttal.  I have taken this into consideration.)

I warmly recommend the authors to integrate the paper with all the useful suggestions received during the review stage.

(Concerning the "interpretability" debate: my take is that the authors seem to be thinking of faithfulness, while the reviewers are thinking of understandability for actual practitioners. If the authors want to prevent further confusion, they'd better clarify what they mean by "interpretability" in the paper, or to avoid the term altogether.)